# Glassy adhesion dynamics govern transitions between sub-diffusive and super-diffusive cancer cell migration on viscoelastic substrates

Vivek Sharma [1,2,10], Kolade Adebowale[3,4,5,10], Ze Gong [6,10], Ovijit Chaudhuri [7,8] & Vivek B. Shenoy [1,2,9]

Cell migration is pivotal in cancer metastasis, where cells navigate the extracellular matrix (ECM) and invade distant tissues. While the ECM is viscoelastic, exhibiting time-dependent stress relaxation, its influence on cell migration remains poorly understood. Here, we employ an integrated experimental and modeling approach to investigate filopodial cancer cell migration on viscoelastic substrates and uncover a striking transition from sub-diffusive to super-diffusive behavior driven by the substrate's viscous relaxation timescale. Conventional motor-clutch based migration models fail to capture these anomalous migration modes, as they overlook the complex adhesion dynamics shaped by broad distribution of adhesion lifetimes. To address this, we develop a glassy motor-clutch model that incorporates the rugged energy landscape of adhesion clusters, where multiple metastable states yield long-tailed adhesion timescales. Our model reveals that migration dynamics are governed by the interplay between cellular and substrate timescales: slow-relaxing substrates prolong trapping, leading to sub-diffusion, while fast-relaxing substrates promote larger steps limiting trapping, leading to super-diffusion. Additionally, we uncover the role of actin polymerization and contractility in modulating adhesion dynamics and driving anomalous migration. These findings establish a mechanistic framework linking substrate viscoelasticity to cell motility, with implications for metastasis and cancer progression.

Cancer cell migration is a fundamental process in tumor progression and metastasis, facilitating cancer cells to traverse the extracellular matrix (ECM), invade surrounding tissues, and establish secondary tumors. This multifaceted phenomenon is modulated by the mechanical properties of both the cells and their ECM, which provide critical mechanical and biochemical cues[1–3]. Among these, the viscoelastic characteristics of the ECM, encompassing elasticity (resistance to deformation) and viscosity (energy dissipation), play a pivotal role in influencing migration phenotypes such as amoeboid, mesenchymal, and filopodial migration[4,5]. The viscoelastic properties of the ECM regulate cell migration through time-dependent stress relaxation and deformation[6,7]. These dynamic mechanical properties modulate cellular processes, including spreading, division, and motility[3,7–9]. While the role of elasticity in migration has been extensively investigated, the impact of viscous dissipation and stress relaxation on migration dynamics remains insufficiently explored. This knowledge gap is

particularly relevant for understanding anomalous diffusion behaviors frequently observed in active cell migration.

Our previous work[7] demonstrated that substrate viscosity and stress relaxation critically influence cell migration dynamics, underscoring the significant regulatory role of ECM viscoelasticity on cellular motility. Nevertheless, this earlier study primarily focused on the phenomenological correlation between viscosity-induced changes in actin cytoskeletal dynamics and overall migration behavior, without providing mechanistic insights into the underlying diffusion mechanism and link to the adhesion level processes. Specifically, we did not investigate how ECM viscoelasticity modulates migration diffusivity characterized through mean squared displacement (MSD), which scales with time as $MSD = \mu t^{\alpha}$, where the exponent $\alpha$ characterizes the diffusivity of cell migration, while $\mu$ represents the diffusion constant. Traditional motor-clutch models, while valuable in studying mechanosensitive responses during migration, typically predict changes only in the diffusion constant ($\mu$) and consistently yield purely diffusive behavior ($\alpha = 1$) migration[10]. However, cell migration is inherently an active process that frequently displays anomalous diffusion ($\alpha \neq 1$), with purely diffusive migration an exception rather than the norm[11,12]. Super-diffusive motion ($\alpha > 1$) is linked to persistent, directed migration, while sub-diffusive motion ($\alpha < 1$) arises from intermittent trapping and saltatory movement.

To uncover the physical mechanisms governing these anomalous cell migration patterns, we develop a glassy motor-clutch model that accounts for the heterogeneous dissociation kinetics of adhesion proteins. The conformational flexibility and mechanical interactions within adhesion complex proteins (e.g., integrins, talin, vinculin), results in a broad distribution of dissociation timescales[3,6-9,13]. This produces a spectrum of transient and prolonged dissociation events in an adhesion cluster, mimicking the hallmark feature of glassy systems[6,14,15]. On viscoelastic substrates, these dissociation dynamics are further modulated by fluctuating mechanical loads and the substrate's time-dependent stress relaxation. Here, we present a physics-based model that unifies sub-diffusive and super-diffusive migration within a single biophysical framework by linking adhesion glassiness to the underlying non-Gaussian trapping-time and step-size statistics. The model integrates glassy adhesion dynamics with substrate viscoelasticity, revealing how the interplay between cellular and matrix timescales governs trapping, step sizes, and the transitions between distinct diffusive regimes. We further validate our model through experiments, demonstrating how substrate stress relaxation modulates migration behavior.

## Results

### Enhanced substrate relaxation promotes the transition from sub-diffusive to super-diffusive migration modes

We first examined how substrate viscoelasticity influences cell migration diffusivity. Prior studies have used tunable viscoelastic hydrogels to investigate migration dynamics[7,16]. Here, we employed interpenetrating network (IPN) hydrogels composed of alginate and reconstituted basement membrane (rBM), where alginate controls mechanical properties, and rBM provides adhesion sites. By adjusting alginate molecular weight, we systematically modulated substrate stress relaxation timescale while maintaining a constant Young's modulus (~2 kPa), consistent with soft tissues such as brain and liver[6]. Human fibrosarcoma HT-1080 cells were cultured on IPNs with low- and high-molecular-weight alginate (Fig. 1A), yielding fast-relaxing (~10 s) and slow-relaxing (~1000 s) substrates, respectively (Fig. 1B). By maintaining a constant Young's and independently modulating substrate stress relaxation timescale, we isolated specific effect of stress relaxation on migration behavior.

To assess migration phenotypes, we conducted time-lapse microscopy experiments and characterized cell motility using the diffusivity exponent $\alpha$ obtained by fitting $MSD \propto t^{\alpha}$ (details of experiments and analysis are provided in the Methods section). For classic Brownian motion $\alpha = 1$, indicating purely diffusive behavior. However, we observed a striking transition in migration dynamics: cells on fast-relaxing substrates displayed super-diffusive migration ($\alpha > 1$), while cells on slow-relaxing substrates showed sub-diffusive migration ($\alpha < 0.8$) (Fig. 1C, D). These findings suggest that substrate stress relaxation independently modulates the transition between super- and sub-diffusive migration.

To further quantify migration persistence, we analyzed two independent metrics: track straightness and velocity autocorrelation (VAC)[17,18]. Track straightness measures the directionality of a migration path as the ratio of net displacement to total distance traveled, while VAC captures how consistently a cell maintains its direction over time, with slower decay indicating greater persistence. Cells on fast-relaxing substrates exhibit greater track straightness than their slow-relaxing counterparts (Fig. 1E), indicating cells have more directed migration on fast-relaxing substrates. Similarly, VAC analysis revealed a slower decay for cells on fast-relaxing substrates, reflecting prolonged persistence times (Fig. 1F). Despite differences in decay rate, all VAC curves plateaued to zero at long timescales, suggesting that over extended durations, migration ultimately transitions to a diffusive regime. Together, these three independent metrics—MSD scaling, track straightness, and VAC—demonstrate that enhanced substrate stress relaxation promotes persistent, super-diffusive migration, while slow-relaxing substrates favor trapped, sub-diffusive behavior.

### Conventional models with a single bond timescale cannot capture anomalous migration modes

We first examined how substrate stress relaxation timescales influence migration modes using a conventional motor–clutch modeling approach. In this model, molecular clutches, which represent adhesion clusters, dynamically link actin filaments to the substrate (Fig. 2A). Clutches are modeled as Hookean springs that randomly engage and disengage, with engagement occurring at a constant association rate and disengagement governed by a force-dependent dissociation rate, as described by Bell's law ($r_{off} = \frac{1}{\tau_{off}} e^{\frac{F_c}{F_b}}$). Forces exerted by engaged clutches counteract actomyosin-driven retrograde flow, allowing actin polymerization at the cell front and propelling forward cell migration. The total force transmitted to the substrate thus emerges stochastically from the binding and unbinding dynamics of multiple clutches.

To simulate the impact of stress relaxation timescales, we vary the substrate viscosity ($\eta$) while maintaining additional stiffness ($k_a$) and long-term stiffnesses ($k_l$) constant, mirroring our experimental approach of altering alginate molecular weight ratios to modify hydrogel stress relaxation while keeping elastic properties unchanged. The stress relaxation timescale in the simulations is expressed as $\frac{\eta}{k_a}$, where a high value of $\eta$ signifies a slow-relaxing substrate and a low value of $\eta$ signifies a fast-relaxing substrate (see Methods for governing equations and simulation details).

Using the conventional motor–clutch model, we found that although cells migrate farther on fast-relaxing substrates (Fig. 1G-i), the model consistently predicts purely diffusive migration ($\alpha \approx 1$), irrespective of the substrate relaxation timescale (Fig. 1G-ii). Thus, the conventional model fails to capture the experimentally observed transition between sub-diffusive ($\alpha < 1$) and super-diffusive ($\alpha > 1$) migration modes. Furthermore, metrics for migration persistence, including track straightness and velocity autocorrelation (VAC), showed no significant differences between slow and fast-relaxing substrates using this conventional approach (Fig. 1G-iii, iv).

These observations underscore the missing physics from the conventional models and the need for capturing long-tailed processes in order to capture the anomaly in migration modes.

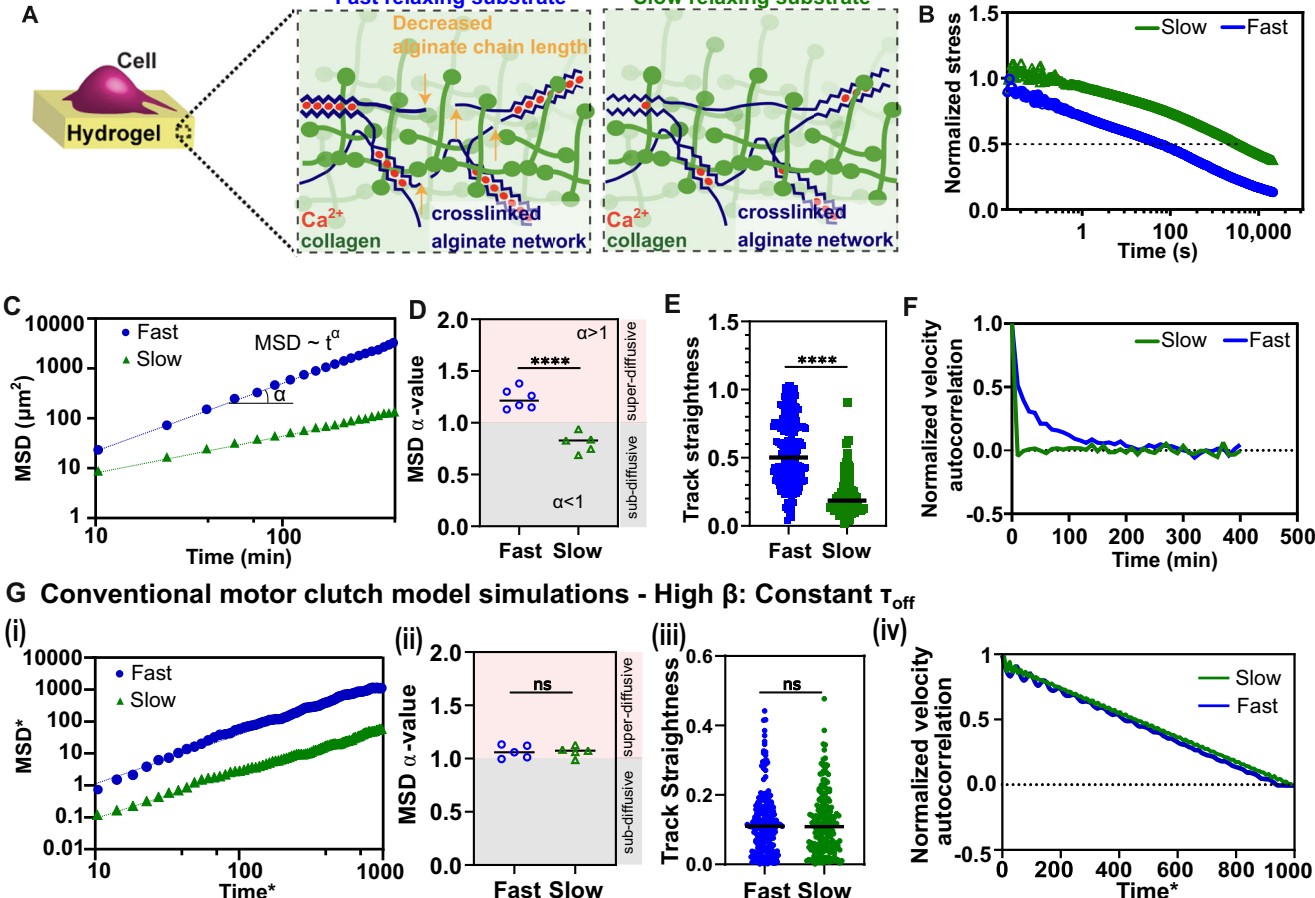

**Fig. 1 | Transition between super- and sub-diffusive cell migration is driven by substrate stress relaxation and cannot be explained by conventional motor clutch models. A** Schematic representation of a cancer cell seeded on an alginate-rBM IPN gel. **B** Normalized stress curve for fast and slow-relaxing IPNs. **C** Mean squared displacement (MSD) for fast and slow-relaxing IPNs (each data point represents the average of multiple cells measured on a single gel). **D** MSD $\alpha$ value showing super-diffusive migration for fast-relaxing substrates and sub-diffusive migration for slow-relaxing substrates. The unpaired two-tailed Student's $t$ test was used for data analysis: $P = 5.13E{-}05$; $n = 6$ (fast) and 5 (slow) fits/experiments, each for over 1000 cells. **E**, **F** Track straightness and Normalized velocity autocorrelation showing higher persistency for fast relaxing substrates compared to slow relaxing substrates. In **E**, the unpaired two-tailed Student's $t$ test was used for data analysis: $P = 2.09E{-}21$; $n = 203$ (fast) and 87 (slow) over three independent samples.

**G** Constant $\tau_{off}$ model. (i) MSD-time fit shows similar behavior/slope obtained from conventional model simulations for both fast and slow relaxing substrates (non-dimensionalised using time scale $1/r_{on}$ and length scale $v_p/r_{on}$). (ii) Diffusivity exponent values confirm that the conventional models capture only purely diffusive modes. The unpaired two-tailed Student's $t$ test was used for data analysis: $P = 0.94$; $n = 5$ (fast) and 5 (slow) fits, each for 100 independent simulations. (iii) The model with a constant off-rate fails to capture the sub-diffusive migration mode and the differences in migration persistence. The unpaired two-tailed Student's $t$ test was used for data analysis: $P = 0.72$; $n = 200$ (fast) and 200 (slow) independent simulations. (iv) Normalized velocity autocorrelation (VAC) function fails to capture the exponential decay of velocity due to the absence of distributed trapping times and step-sizes. (Black lines represent mean in the violin plots. 'ns' represents not significant $p > 0.05$, *$p < 0.05$, **$p < 0.01$, ***$p < 0.001$, ****$p < 0.0001$).

## The glassy motor clutch model predicts both super- and sub-diffusive migration modes

To overcome the limitations of the conventional model, we developed a glassy motor–clutch framework (Fig. 2A) that incorporates a broad distribution of adhesion lifetimes. Unlike previous models that assume a single characteristic off-rate ($\tau_{off}$), our approach accounts for the rugged energy landscape of adhesion clusters (Fig. 2B), where adhesion proteins undergo stochastic unfolding and unbinding with highly variable kinetics[19–21]. This behavior is characteristic of glassy systems and gives rise to long-tailed dissociation time distributions (see Methods).

To capture the glassy adhesion dynamics, we adopt a power law distribution of $\tau_{off}$, expressed as $p(\tau_{off}) = \frac{|(\beta-1)|}{\tau_{min}} \left(\frac{\tau_{off}}{\tau_{min}}\right)^{-\beta}$, where $\beta$ and $\tau_{min}$ represent the glass coefficient and characteristic dissociation time constant[22,23]. This distribution enables us to capture both frequent, short-lived and rare, long-lived adhesion events, which are crucial for modeling the heterogeneity of adhesion cluster behavior. The shape of

this distribution is shown in Fig. 2C, and its cumulative form is depicted in Fig. 2D. When $\beta < 3$ the central limit theorem breaks and the variance of the distribution diverges, leading to non-Gaussian dynamics that are central to reproducing anomalous migration modes (see Methods for details).

Using this glassy model, we simulated cell migration on fast- and slow-relaxing substrates. The model successfully predicts that fast-relaxing substrates promote increased cell migration distance with an MSD power law coefficient ($\alpha > 1$) leading to super-diffusive migration, whereas cells on slow-relaxing substrates exhibit sub-diffusive migration ($\alpha < 1$) (Fig. 2F-i, ii). Consequently, enhanced substrate relaxation resulting from decreased viscosity induces a transition from a sub-diffusive to a super-diffusive migration mode. To quantitatively examine this transition, we evaluate migration characteristics through track straightness and VAC. We find that cells on fast-relaxing substrates display greater migration track straightness and a gradually decaying VAC, indicating greater persistency in migration. On the other hand, we find that cells on slow-relaxing substrates display lower

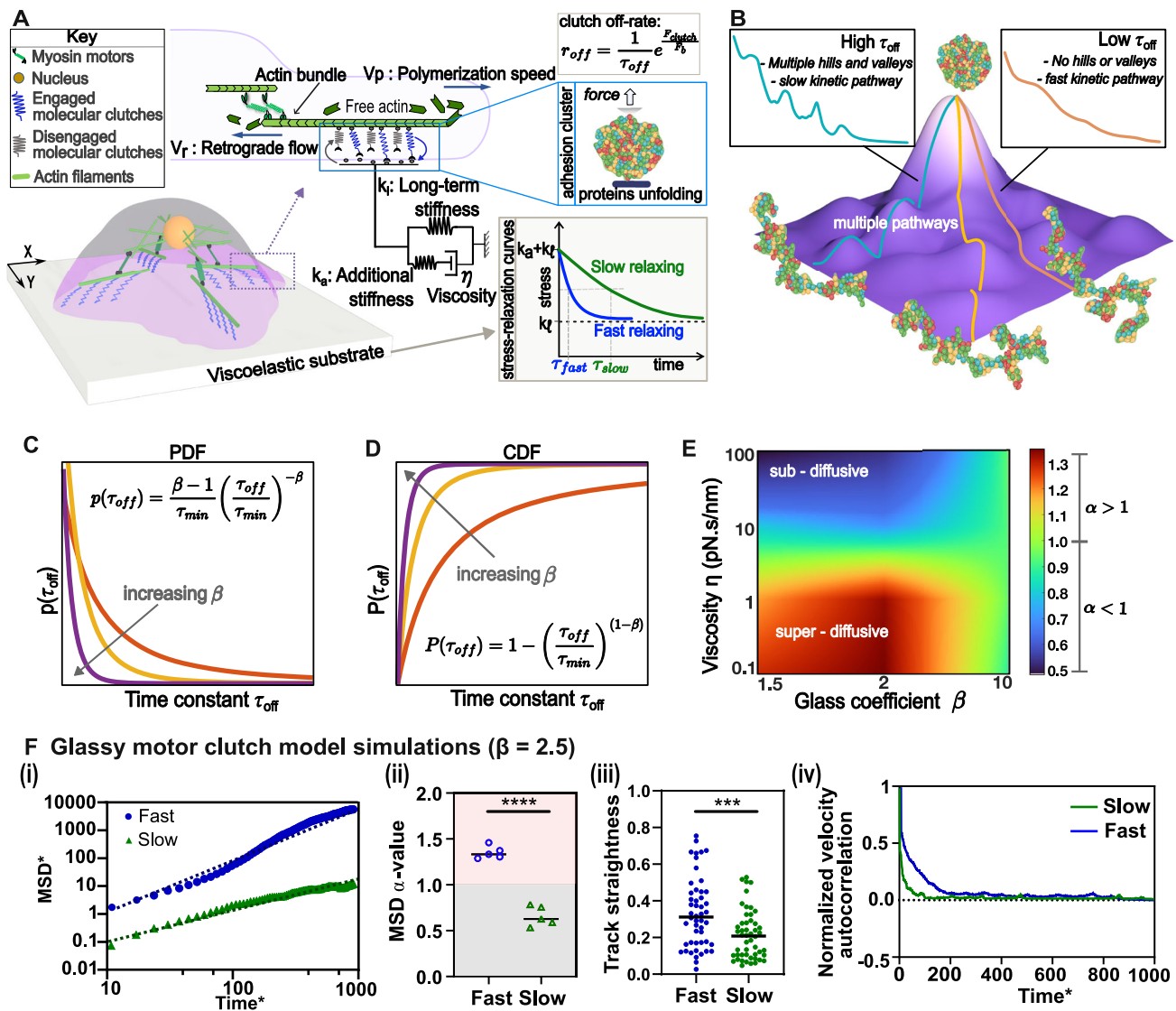

**Fig. 2 | The glassy motor-clutch model captures both sub- and super-diffusive migration modes modulated by substrate stress relaxation. A** Schematic representation of the 2D motor clutch model illustrating cell migration on a viscoelastic substrate. The model utilizes independent clutch modules in the X and Y directions. **B** Energy landscape depicting protein unfolding with multiple pathways, each with varying off rates. **C** Probability density function (PDF) for the off-rate time constant with varying glass coefficient ($\beta = 1.5, 3, 10$). **D** Cumulative distribution function (CDF) of the power law distribution used for sampling off rates ($\beta = 1.5, 3, 10$). **E** Heatmap illustrating variation in migration diffusivity parameter ($\alpha$) with viscosity ($\eta$) and glass coefficient ($\beta$). Higher values of $\beta$ fail to capture both sub- and super-diffusive regimes, underscoring the importance of long-tails in distribution. **F** Glassy motor clutch model. (i) MSD time curves show that cells on fast-relaxing substrates tend to travel longer distances compared to their slow-

relaxing counterparts and consequently have a higher slope. (ii) Diffusivity exponent ($\alpha$) values representing the diffusive mode. Cells on slow-relaxing substrates exhibit sub-diffusive behavior, while those on fast-relaxing substrates display super-diffusive behavior. The unpaired two-tailed Student's $t$ test was used for data analysis: $P = 2.01E{-}06$; $n = 5$ (fast) and 5 (slow) fits, each for 100 independent simulations. (iii) Glassy model captures trends in track straightness and VAC. Enhanced substrate relaxation promotes directional migration, reflected in the higher velocity autocorrelation observed for cells on fast-relaxing substrates. The unpaired two-tailed Student's t-test was used for data analysis: $P = 0.00048$; $n = 50$ (fast) and 50 (slow) independent cell trajectory simulations. (iv) Normalized VAC function decays more slowly for fast-relaxing substrates compared to medium- or slow-relaxing substrates. (Black lines represent mean in the violin plots. 'ns' represents not significant $p > 0.05$, *$p < 0.05$, **$p < 0.01$, ***$p < 0.001$, ****$p < 0.0001$).

migration track straightness and a fast-decaying VAC (Fig. 2F-iii, iv), consistent with experimental observations (Fig. 1), reinforcing a lower persistency in migration behavior. Importantly, both sub- and super-diffusive modes of cell migration are captured only when the glass exponent for the clutch off-rate, $\beta$, is less than or approximately equal to 3 (Fig. 2E, detailed trajectories on this phase diagram shown in Supplementary Fig. 4). This range of $\beta$ is critical because it allows the distribution of off-rates to exhibit long-tailed behavior, which is a defining characteristic of glassy clutch dynamics. In this regime, the off-rate distribution is heavily skewed toward larger timescales, meaning that adhesion bonds exhibit a broad spectrum of dissociation

times, including rare but significantly slow dissociation events. These long tails are essential for representing the diverse relaxation time-scales that underlie complex cell migration dynamics on viscoelastic substrates.

Long-tailed distributions are characterized by their influence on the variance of the off-rate constant. As discussed in the last section, for $\beta > 3$, the variance of the off-rate distribution becomes finite, leading to a more predictable, Gaussian-like behavior. However, for $\beta < 3$, the variance becomes infinite, which breaks the central limit theorem and leads to a non-Gaussian distribution of dissociation events (see Supplementary Note 1). This variability in the off-rate

constant plays a significant role in shaping cell migration trajectories, introducing non-uniformity of each migration step, as observed experimentally. In the following sections, we explore these trajectories in detail, examining how variations in substrate relaxation timescales change their dynamics and eventually contribute to the emergence of sub- and super-diffusive migration modes.

## Interplay between cell trapping time and migration step size drives the transition between sub- and super-diffusive migration modes

To gain mechanistic insights into the transition between sub-diffusive and super-diffusive cell migration, we performed a detailed analysis of individual cell trajectories. In the model, cell migration dynamics are governed by both the waiting times between steps and the magnitude of the steps themselves. To better understand these waiting times and step sizes - and how they vary with changes in stress-relaxation, we focus on a simplified one-dimensional model (Fig. 3A). This reduction preserves the essential physics of the system while enabling a clearer, more focused analysis of individual migration trajectories over time. While the clutches on two ends may initially be balanced (Fig. 3A-i), the stochastic nature of adhesion bond dynamics leads to an imbalance in the number of bound clutches. Consequently, one end harbors fewer bound clutches, causing each bound clutch to bear a greater level of force, as the total force must be equal and opposite on both ends. As the clutches begin to break, the load shared among the bound clutches increases, increasing the likelihood of breakage due to the force dependence of the off rate, $r_{off}$. Ultimately, all the clutches on one end undergo catastrophic failure, leading to the disintegration of the entire adhesion cluster (Fig. 3A-ii). The retrograde velocity is greater at the end where more clutches fail (as the lower resistance), which propels the cell toward the opposite end with a velocity of $D_s = V_m \Delta t$ (Fig. 3A-iii). However, before the cell takes its next step in a similar fashion, there is an intermediate period where clutches remain balanced on both ends, as is evident from the distribution of bound clutches (Fig. 3B-i). We term this duration of minimal net displacement the "trapping time" (Fig. 3A-iii, iv). The trapping time is governed by the net dissociation rate $r_{off}$, which is dependent on two factors: the glassy dissociation rate constant $\tau_{off}$ and the clutch forces $F_c$. These two factors are dictated by the substrate relaxation timescale. Eventually, the cell emerges from the trapped state upon experiencing a force imbalance between the two ends (Fig. 3A-v). Next, we examine cell trajectories on slow- and fast-relaxing substrates, revealing distinct migration patterns (Fig. 3B). On slow-relaxing substrates, cells undergoing sub-diffusive migration display balanced clutches at both ends for extended durations (Fig. 3B: left panel). Consequently, their migration distance profiles exhibit prolonged periods of nearly zero displacement, indicating longer trapping times $t_p$. The migration velocity curve confirms this trapped state, showing near-zero migration velocity, signifying balanced retrograde flow velocities at both ends. Conversely, on fast-relaxing substrates, the bound clutch probability, migration distance, and velocity plots (Fig. 3B: right panel) illustrate that clutches at the two ends do not remain balanced for prolonged durations, thus limiting the occurrence of extended trapping times.

We then quantified the observed differences in trapping times and step sizes and found that our model predicts significantly longer trapping times on slow-relaxing substrates compared to fast-relaxing substrates (Fig. 3Ei). Further, the model predicts larger migration step-sizes for cells on fast-relaxing substrates in contrast to those on slow-relaxing substrates (Fig. 3Eii). Experiments validated that cells exhibit intermittent trapping between consecutive migration steps, displaying a discontinuous migration pattern (i.e., each migration step is different during the whole trajectory of a cell) (Fig. 3C, D). Quantification of these experimental trapping times confirmed that, on average, cells on slow-relaxing substrates experienced significantly longer trapping

times than did their counterparts on fast-relaxing substrates (Fig. 3Fi), in agreement with theoretical predictions. Further experimental quantification of step-sizes validated the presence of larger steps taken by cells on fast-relaxing substrates (Fig. 3Fii).

In addition to differences in the magnitude of trapping times, a defining feature of sub-diffusive migration on slow-relaxing substrates is the emergence of long-tailed trapping-time distributions. Likewise, for cells exhibiting super-diffusive behavior on fast-relaxing substrates, step sizes follow a heavy-tailed distribution, and these statistics are central to the onset of anomalous migration. Sub-diffusive motion is often described using continuous-time random walk (CTRW) or fractional diffusion models[11,24,25], in which the waiting time between steps follows a power-law distribution with long tails, unlike standard random walks that assume uniform waiting times. The presence of rare but prolonged trapping events lowers the overall displacement, thereby reducing migration diffusivity. This same mechanism, driven by long-tailed trapping-time statistics, underlies sub-diffusivity in our glassy motor–clutch model.

Similarly, super-diffusive motion is typically represented by the Lévy walk model[11,25], where step lengths are drawn from a heavy-tailed distribution to produce occasional long jumps. In our framework, such heavy-tailed step-size behavior is not imposed but emerges naturally from glassy adhesion dynamics and the broad distribution of clutch off-times. These rare, large steps are an intrinsic outcome of the coupling between adhesion kinetics and substrate relaxation. Importantly, CTRW, fractional diffusion, and Lévy walk models are phenomenological: they assume the existence of heavy-tailed statistics but do not explain their physical origin. In contrast, our glassy motor–clutch model is physics-based. The trapping times and step-size distributions arise self-consistently from force transduction and adhesion dynamics that respond to substrate stress relaxation. This mechanistic foundation distinguishes our framework from purely statistical descriptions and reveals how anomalous migration emerges directly from the underlying biophysical processes.

To validate the role of heavy tails for step-size and trap-time distribution as the governing physics controlling diffusivity, we utilize the Kurtosis number (Fig. 3G). The Kurtosis number is a statistical measure that quantifies the "tailed-ness" of a probability distribution compared to a normal distribution. It is calculated using the fourth standardized moment of the distribution and indicates whether the data has heavier, or lighter tails compared to a normal distribution. In this way, the relative contribution of step-size distribution tailed-ness $K_{step}$ to the trapping time distribution tailed-ness $K_{trap}$ can be compared, by taking a ratio of their respective Kurtosis numbers. We then use this ratio to compare how viscosity and glass coefficient affect the tailed-ness of both step size and trapping time distributions and consequently migration diffusivity. We note that for low values of glass coefficient $\beta$, the ratio $K_{trap}/K_{step}$ increases as the viscosity increases, and for high values of $\beta$, the substrate viscosity has negligible effect on the Kurtosis ratio (Fig. 3H). Interestingly, comparing the phase map previously obtained for migration diffusivity (Fig. 2E) with the phase map obtained for the Kurtosis number ratio (Fig. 3H), we find overlapping trends. The region with large $K_{trap}/K_{step}$ in Fig. 3H overlaps closely with the sub-diffusive region in Fig. 2E, demonstrating that sub-diffusive migration occurs when the relative contribution of $K_{trap}$ is large, effectively making $K_{trap}/K_{step}$ large. This underscores the importance of a long tail for the trap-time distribution in capturing sub-diffusive migration. Conversely, the region where $K_{trap}/K_{step}$ is small in Fig. 3H, we observe a correspondence with the super-diffusive region in Fig. 2E. In this case, the relative contribution of $K_{step}$ is large, effectively making $K_{trap}/K_{step}$ relatively smaller. This observation of increasing tailedness for step-size distribution highlights the significance of long tails in the step-size distribution in generating super-diffusive migration. Both our experiments and simulations for fast- and slow-relaxing substrates confirm these trends (as seen in Fig. 3E, F),

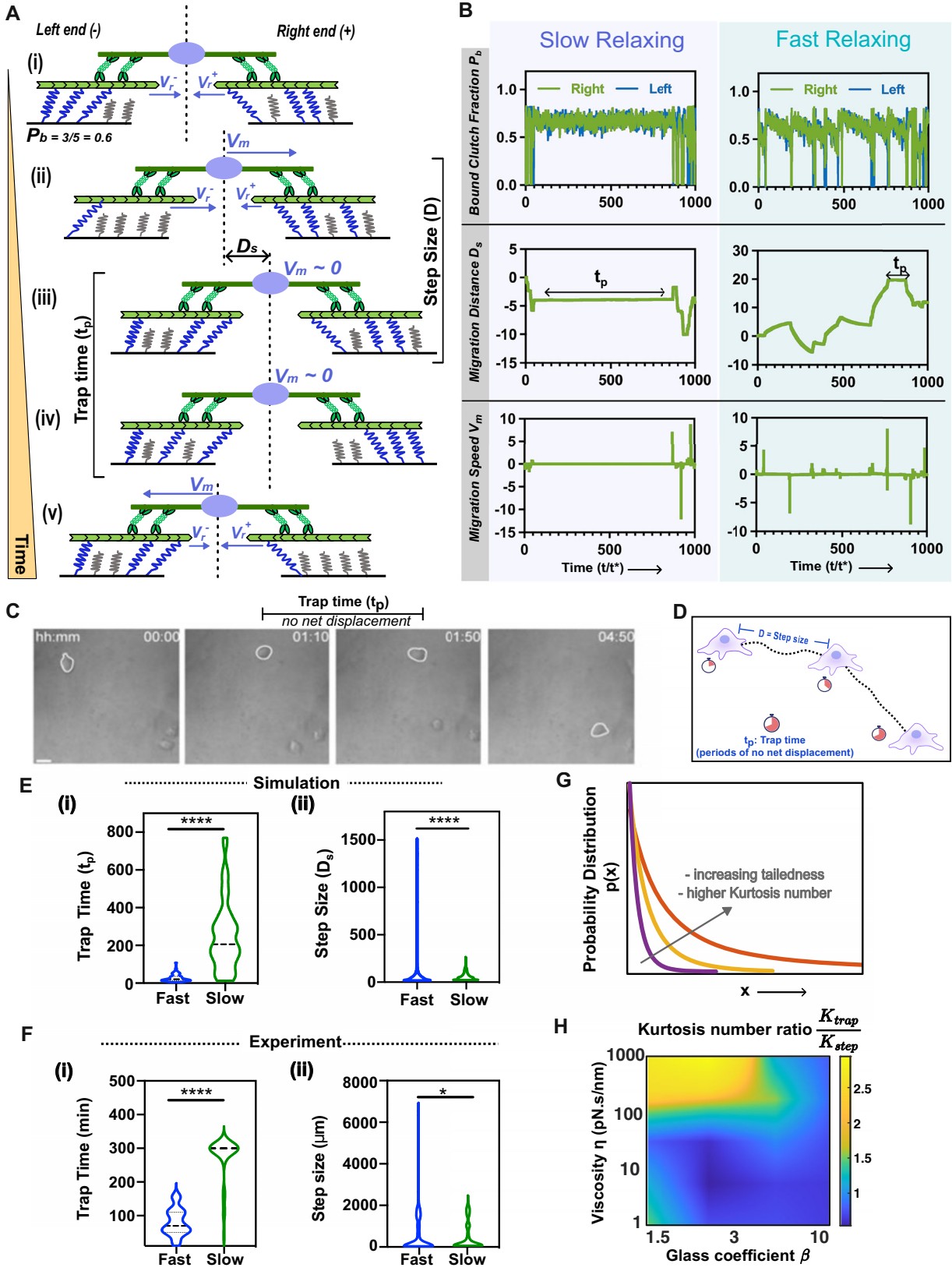

where cells on slow-relaxing substrates showing sub-diffusivity have heavy-tailed trapping times while cells on fast-relaxing substrates showing super-diffusivity have heavy-tailed step sizes. Thus, we conclude that migration diffusivity is regulated by the substrate stress relaxation via the competition between trap time, step sizes and their respective distributions.

The inability of conventional motor–clutch models to reproduce these trapping statistics is further demonstrated in Supplementary Figs. 6, 7. As shown in Supplementary Fig. 6, trajectories generated by the standard model with constant clutch off-times produce uniform, Gaussian migration with narrowly distributed trapping durations. Even when model parameters such as clutch number, motor number, or stall

**Fig. 3 | Simulations implicate longer trapping time gives rise to sub-diffusive migration and larger step sizes gives rise to super-diffusive migration.**
**A** Sequential depiction of cell-matrix interactions during migration: (i) Clutch imbalance at the two ends governs migration distance $D_s$ and speed $V_m$. (ii) Clutch failure on the left end initiates cell movement to the right. (iii)-(iv) Migration occurs intermittently, with pauses (referred to as trap time) due to clutch balance. (v) Clutch failure on right end propels cell to the right. **B** Bound clutch fraction shows the clutch distribution at the two ends. Migration distance and velocity plots represent the migration trajectory with time. Clutches remain balanced during periods of trapping and correlate with minimal change in migration distance and speed. **C** Experimental images illustrate periods of cell trapping with no net displacements during migration trajectory. **D** Schematic representation of trapping parameters and step size. **E**, (i) Simulations predict longer trapping times with long-tails for slow-relaxing substrates and (ii) large step sizes with heavy-tailed distribution on fast-relaxing substrates. **F** Experiments validate trap time and step size distributions for slow and fast-relaxing substrates. The unpaired two-tailed Student's $t$ test was used for data analysis. **E**(i): $P = 1.52E{-}13$; $n = 50$ (fast) and 50 (slow) independent simulations; **E**(ii): $P = 1.55E{-}39$; $n = 1383$ (fast) and 1771 (slow) observations across 50 independent trajectory simulations; **F**(i): $P = 1.79E{-}21$; $n = 36$ (fast) and 32 (slow) cells over three experiments; **F**(ii): $P = 0.039$; $n = 125$ (fast) and 88 (slow) cells over three different experiments. **G** Kurtosis number captures heavy-tailedness. For a distribution having rare events, the Kurtosis number is higher. **H** The Kurtosis number ratio between trapping time and step size is correlated with the sub-diffusive migration mode. (Black dotted lines represent mean in the violin plots. 'ns' represents not significant $p > 0.05$, *$p < 0.05$, **$p < 0.01$, ***$p < 0.001$, ****$p < 0.0001$).

duration are systematically varied (Supplementary Fig. 7), the resulting trajectories remain diffusive, showing only changes in average speed or pause length but never exhibiting the long-tailed, non-Gaussian statistics characteristic of anomalous migration. These results confirm that parameter tuning within conventional formulations cannot generate intermittent trapping or anomalous diffusion and that such behaviors emerge only when adhesion glassiness is explicitly incorporated into the model.

## Trap time and step size are regulated by the interplay of substrate relaxation timescales and glassy adhesion dynamics

To understand the mechanisms that regulate trapping times and step sizes in cell migration, we identify three critical timescales governing adhesion dynamics, force generation, and viscoelastic substrate relaxation. These timescales determine whether migration will manifest as sub-diffusive with prolonged trapping periods, or super-diffusive featuring large step sizes.

1. Motor-stall timescale ($\tau_l$): Myosin motors generate contractile forces as they pull actin filaments against cellular load. This force is regulated by Hill's relation, where the motor's velocity declines with increasing load until it reaches a "stall" state. This "stall force" represents the maximum load at which the motor can no longer perform mechanical work against resistance, and at this point, its velocity approaches zero. The timescale for this process is given by Gong et al.[3], $\tau_l = \frac{F_m N_m}{v_u k_l}$ (-10 s), where $F_m$ represents the individual motor's stall force, $N_m$ the number of motors, $v_u$ the unloaded motor velocity, and $k_l$ the long-term stiffness. This timescale, $\tau_l$ defines the period required for motors to reach the stall state and it modulates the dynamics of force transmission.

2. Dissociation timescale ($\tau_{off}$): Adhesion clusters modeled as molecular clutches serve as the link that transmit actomyosin generated force to the substrate. These clutches can associate (taken as a constant rate) and dissociate with a characteristic dissociation timescale $\tau_{off}$ at every migration step. We account for the glassy dynamics of the adhesion cluster protein by sampling $\tau_{off}$ from a power-law distribution $p(\tau_{off}) = \frac{|(\beta-1)|}{\tau_{min}}\left(\frac{\tau_{off}}{\tau_{min}}\right)^{-\beta}$ as discussed in the earlier sections (range: -1–100 s). This timescale regulates the intrinsic tendency of clutches to stay bound and captures the variability in migration at each step.

3. Substrate relaxation timescale ($\tau_s$): Once the force is transmitted to the substrate through an adhesion cluster, the viscoelastic substrate undergoes stress relaxation. This response is characterized by a timescale, $\tau_s = \eta/k_a$, where $\eta$ represents the viscosity and $k_a$ is the additional stiffness of the substrate. Stress relaxation reduces the effective force in the clutches over time, thus modulating the adhesion dynamics differently for fast-($\tau_s \sim 10s$) and slow-relaxing substrates ($\tau_s \sim 1000s$).

The interplay of these three timescales determines whether cell migration shows sub-diffusive (dominated by prolonged trapped periods) or super-diffusive (characterized by large displacement steps). We outline a few relevant cases of how these timescales specifically affect the propensity for a cell to be trapped or show large migration steps.

For fast-relaxing substrates, the timescale for substrate relaxation is faster than the motor-stall timescale ($\tau_s < \tau_l$) and we get the following cases:

(i). $\tau_{off} < \tau_s < \tau_l$ : A low $\tau_{off}$ means that molecular clutches have an intrinsic nature of quickly disengaging, i.e., they remain engaged for a short time. In the case when ($\tau_{off} < \tau_s$) the clutches disengage faster than the substrate can relax to its long-term stiffness $k_l$. This results in quick clutch failure and lower resistance to retrograde flow. Since the overall retrograde flow is higher, cells take large steps and are unable to remain trapped (Fig. 4B-iv).

(ii). $\tau_s < \tau_l < \tau_{off}$ : This is possible only for the glassy motor clutch model in the presence of high $\tau_{off}$ as shown in Fig. 4B. The substrate relaxes to long-term stiffness $k_l$ quickly, and clutches sense a change in stiffness before motors have stalled. The retrograde flow of actin continues to pull on the clutches, resulting in eventual clutch failure (Fig. 4B-iv). In this case, quick relaxation coupled with high retrograde flow causes clutch failure, preventing trapping. Hence, even though clutches are intrinsically stable and can remain engaged for longer, fast substrate relaxation does not let the force build up and resist retrograde flow and hence causes eventual clutch failure (Fig. 4B-iii).

For slow relaxing substrates the substrate takes much longer to relax compared to the motor stall timescale ($\tau_s > \tau_l$) and we get following cases:

(i). $\tau_{off} < \tau_l < \tau_s$ : The intrinsic nature of clutches is to disengage quickly due to low $\tau_{off}$, and since clutches disengage faster than the substrate relaxes or motors are stalled, the cell cannot remain trapped for longer (akin to Fig. 4B, ii).

(ii). $\tau_l < \tau_{off} < \tau_s$ : This is only possible for the glassy distribution of $\tau_{off}$ since it requires sampling of high $\tau_{off}$ values, which can only occur when long tails are present. Clutches remain engaged for long periods because the motors have reached their stall state, intrinsic $\tau_{off}$ is high enough to provide stability, and the substrate is able to maintain stiffness for long time hence sustaining resistance to retrograde flow. Thus, the cell remains trapped for long periods in this case and undergoes a migration step only after a long time (Fig. 4B-i). We get these long trapping period cases for slow-relaxing substrates when $\tau_s$ is high.

To summarize, for fast-relaxing substrates, since the relaxation time $\tau_s$ is the fastest timescale among the three at play, quick clutch failure is initiated for all cases. This leads to lower retrograde flow resistance, causing cells to take larger migration steps and prevent trapping, leading to super-diffusive migration. Conversely, for slow-relaxing substrates, $\tau_s$ is high and allows motors to stall before the

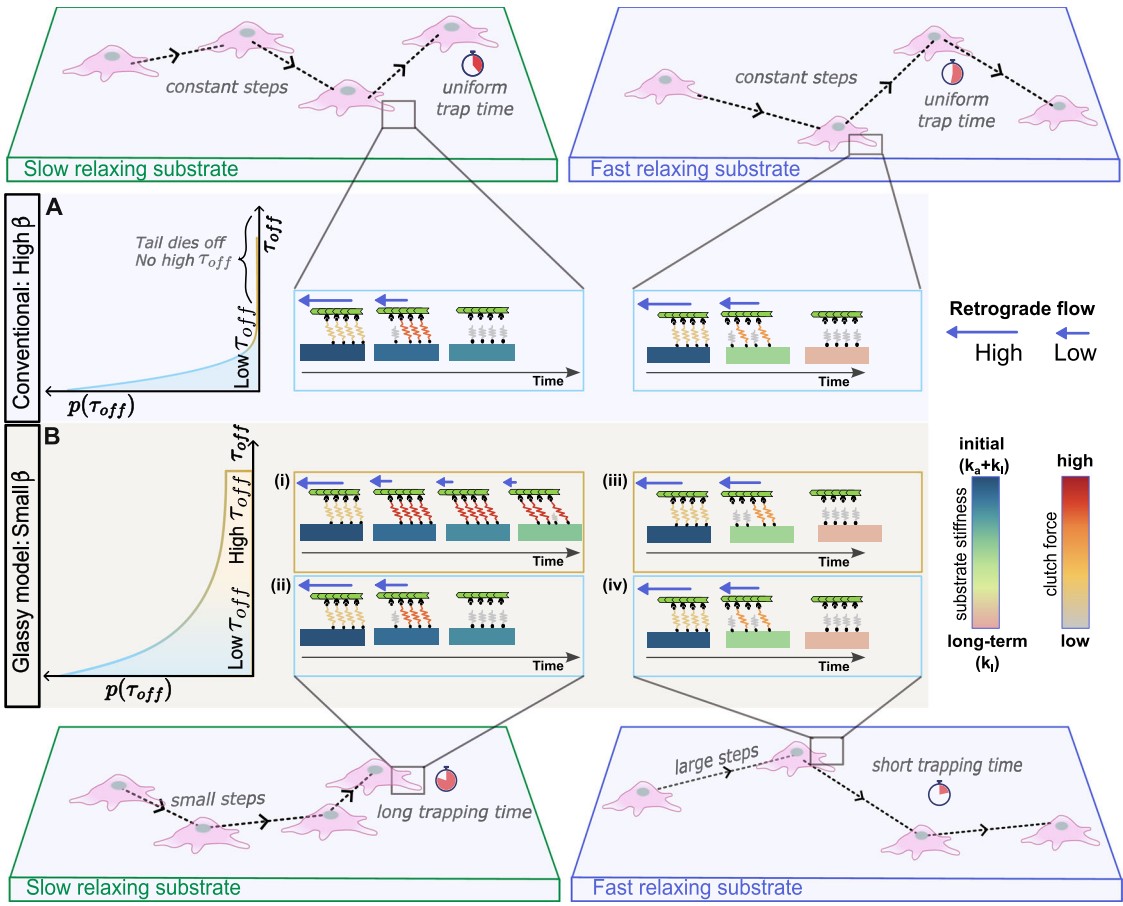

**Fig. 4 | The glassy motor-clutch model mechanistically explains observed differences in migration patterns. A** Clutch dynamics for conventional model (constant $\tau_{off}$c) on fast and slow relaxing substrates with insets showing a similar migration trajectory with uniform distribution of trap time and step sizes. **B** Clutch dynamics with a heavy tailed distribution on fast- and slow- relaxing substrates. Color bar for substrate stiffness shows the relaxation of substrate stiffness from initial stiffness to long term stiffness. Note the time it takes for stiffness change for slow and fast relaxing substrates. Color bar for clutch force shows average force in the adhesions. (i) Clutches stay engaged for long times resulting in cell trapping. (ii) Clutches disengage quickly resulting in short migration steps. (iii) Even for high $\tau_{off}$, clutches disengage quickly on fast-relaxing substrates, preventing any trapping and take large migration steps because of higher retrograde flow. (iv) Quick disengagement of clutches prevents trapping, and the cell shows load and failure dynamics.

substrate relaxes. In this case, whenever a high $\tau_{off}$ is sampled, it provides an opportunity for clutches to stay engaged for longer, resulting in cell trapping zones and hence displaying a sub-diffusive migration mode. It is to be noted that cells do not show a uniform migration pattern for every migration step and rather show a saltatory migration pattern. This behavior is effectively captured in our model by the glassy dynamics of adhesion clusters, which introduces a range of dissociation pathways with a probabilistic distribution of the off-rate time constant $\tau_{off}$ invoked at every migration step. This dynamic interplay of timescales ensure that each migration step is unique, encompassing a spectrum of migration steps and trajectories while predicting the overall migration pattern accurately for both sub- and super-diffusive migration mode. In the next section, we discuss how changes in the active forces generated by actomyosin contractility affect the related timescales and eventually the migration mode since for a cell to show super-diffusive migration, it needs active cellular forces to generate traction and avoid getting trapped.

### Glassy motor clutch model reveals super-diffusion necessitates active process and inhibiting actomyosin activity results in sub-diffusive migration

In our model, clutches serve as dynamic linkages between cells and substrates, constraining movement until sufficient traction forces break these connections to enable migration. Actin polymerization and myosin-driven contractility generate the cytoskeletal forces required for this process, relying on the hydrolysis of ATP to ADP with the release of inorganic phosphate (Fig. 5J). Unlike passive Brownian motion, cell migration is an inherently active process, necessitating continuous energy expenditure to produce these dynamic forces[11,26]. Super-diffusive migration arises when cells sustain contractility, generating forces to overcome trapping and facilitate the cyclic adhesion remodeling essential for persistent motion. To elucidate the role of actomyosin forces in super-diffusive behavior, we manipulated biophysical parameters representing actin polymerization and myosin activity in our model. These computational predictions were experimentally validated by pharmacologically inhibiting actomyosin activity, confirming its critical influence on the observed migration dynamics.

Inhibiting myosin motor activity by reducing the number of myosin motors (parameter $N_m$) led to a decrease in MSD-α values, indicating a shift from super-diffusive to sub-diffusive migration behaviors on fast-relaxing substrates (Fig. 5A–D). The reduction in number of myosin motors results in decreased actomyosin contractility and the level of force transduction through clutches decreases. Following Bell's law for adhesion bonds, lower clutch force leads to a lower off-rate $r_{off}$, leaving bonds bound for longer durations. As a result, cells experience prolonged trapping times, leading to decreased diffusivity, as explained previously. Therefore, myosin

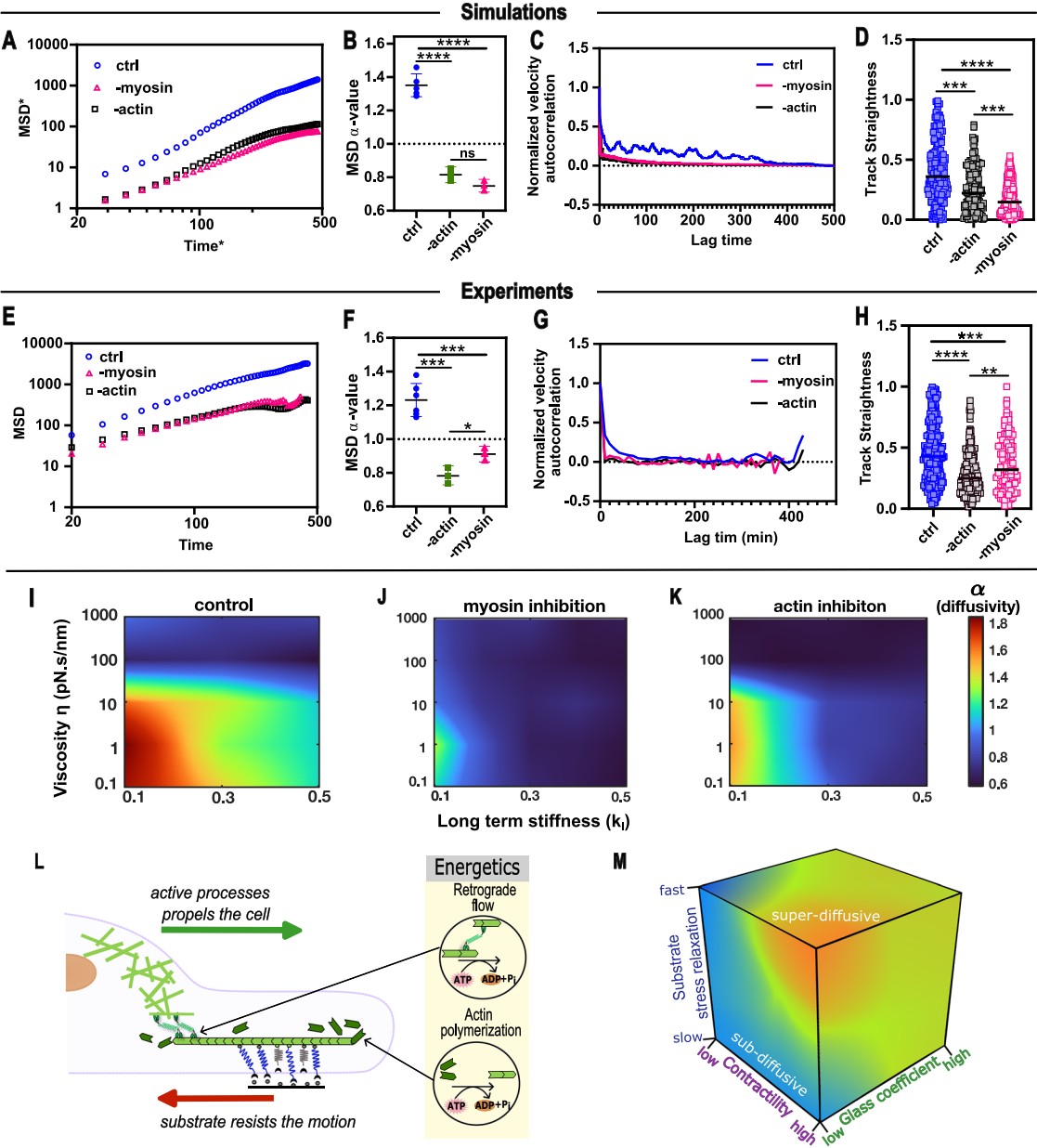

**Fig. 5 | Glassy motor clutch model predicts the transition of super-diffusive to sub-diffusive migration upon actomyosin inhibition. A, B** Simulation inhibition of actin polymerization or myosin II activity analyzed by changing parameters like polymerization speed and number of myosin motors. We observe decreased MSD in both cases with sub-diffusive migration behavior. **C, D** VAC, and track straightness data show that both actin and myosin inhibition reduce migration persistence. **E–H** Experimental inhibition of actin polymerization or myosin II activity decreases MSD, directionality, and persistency migration. The unpaired two-tailed Student's *t* test was used for data analysis. **B** $P = 7.77E{-}06$ (ctrl and -myosin), $1.89E{-}05$ (ctrl and -actin) and $0.07354$ (-actin and -myosin); $n = 5$ (ctrl), 3 (-actin) and 3 (-myosin) fits for 100 independent simulations; **D** $P = 1.00E{-}24$ (ctrl and -myosin), $1.00E{-}11$ (ctrl and -actin) and $3.08E{-}05$ (-actin and -myosin); $n = 200$ (ctrl), 200 (-actin) and 200 (-myosin) independent cell trajectory simulations; **F** $P = 0.00094$ (ctrl and -myosin),

$0.00016$ (ctrl and -actin) and $0.03218$ (-actin and -myosin); $n = 6$ (ctrl), 3 (-actin) and 3 (-myosin) fits/experiments, each for over 1000 cells; **H** $P = 0.00041$ (ctrl and -myosin), $8.03E{-}14$ (ctrl and -actin) and $0.00277$ (-actin and -myosin); $n = 285$ (ctrl), 179 (-actin) and 114 (-myosin) cell trajectories over two independent experiments. **I–K** Heatmaps show a shift toward sub-diffusivity in migration modes upon myosin and actin inhibition for a range of substrate parameters. **L** Schematic showing energy-consuming cytoskeletal processes making cell migration an active process. **M** Schematic phase diagram integrates substrate relaxation, contractility levels, and glass coefficient β, showcasing how the inclusion of glassy dynamics captures the anomalous migration patterns observed across different viscoelastic environments. (Black lines represent mean in the violin plots. 'ns' represents not significant $p > 0.05$, *$p < 0.05$, **$p < 0.01$, ***$p < 0.001$, ****$p < 0.0001$).

activity inhibition increases trapping time due to decreased contractility, thereby inducing sub-diffusive migration. To validate these findings, we inhibited myosin light chain kinase (-MLCK) and blocked MLC phosphorylation, effectively reducing contractility. Consistently, experimental findings mirrored the model predictions, showing a transition from super-diffusive to sub-diffusive migration on fast-relaxing substrates upon MLCK inhibition (Fig. 5E–H). Furthermore,

our model predictions for the effect of MLCK inhibition across a wide range of substrate parameters revealed significant changes in the diffusivity heatmap (Fig. 5I-ii), indicating a predominance of sub-diffusive migration across most parameter regimen.

We similarly simulated actin inhibition by reducing the actin polymerization speed and decreasing the number of myosin motors in the model. Such as for the myosin inhibition study, we observed that

actin inhibition resulted in lower clutch force generation and prolonged clutch binding, leading to longer trapping times and sub-diffusive behavior on fast-relaxing substrates, as evidenced by a reduction in the diffusivity exponent (Fig. 5A–D). To validate these findings, we conducted actin polymerization inhibition studies using Latrunculin A (LatA), a potent toxin that disrupts actin filament polymerization by binding to monomeric actin (G-actin), thereby preventing its incorporation into growing actin filaments (F-actin). Consequently, LatA treatment induces the depolymerization of existing actin filaments and inhibits the assembly of new filaments, thereby reducing polymerization at the leading edge. This disruption of the actin network also weakens the scaffolding needed for myosin II to generate tension, leading to a marked decrease in actomyosin contractility[27,28]. The experimental findings again showed a transition from a super-diffusive to sub-diffusive regime upon actin inhibition. Furthermore, our model analysis of actin inhibition across a broad parameter range revealed a shift toward predominantly sub-diffusive migration, as depicted in the diffusivity heatmap (Fig. 5I-iii).

Reduction in contractility lowers the motor-stall timescale $\tau_l = \frac{F_m N_m}{V_u K_t}$, which is crucial in governing trap time, step sizes, and consequently the migration diffusivity regimes. This causes motors to stall faster, increasing the probability that the dissociation time constant $\tau_{off}$ sampled and the substrate relaxation timescale $\tau_s$ is higher than the motor-stall timescale $\tau_l$, resulting in a stable clutch for longer durations. The elongation of stable clutch durations leads to extended cell trapping and the emergence of sub-diffusive behavior, even on fast-relaxing substrates. This shift in migration mode also correlates with reduced migration persistence, as indicated by trends in velocity autocorrelation (VAC) and track straightness under conditions of myosin and actin inhibition (Fig. 5G, H). Sustaining cell migration requires active energy consumption to drive the cytoskeleton machinery for actin polymerization and actomyosin contractility (Fig. 5L). Diffusivity is therefore directly influenced by the energy available to power the migration machinery: higher diffusivity corresponds to greater active energy expenditure. Inhibition of actomyosin activity reduces energy consumption, leading to decreased cell activity and diminished force generation. Consequently, cells are less able to escape trapped zones, increasing trapping times and further reinforcing sub-diffusive migration.

Based on these observations, our glassy motor clutch model provides a framework to describe migration dynamics across regimes. We propose a phase diagram incorporating substrate stress relaxation, system glassiness (quantified by the glass coefficient β), and contractility levels (Fig. 5M). The inclusion of glassy dynamics is essential for capturing the anomalous migration behaviors observed experimentally and offers insights into the interplay between substrate properties and active cellular processes.

## Discussion

In this study, we developed a comprehensive physics-based model to elucidate how substrate stress relaxation regulates cell migration. Our model reveals that the substrate relaxation timescale dictates transitions between sub-diffusive and super-diffusive migration by modulating trapping and step dynamics. Specifically, slow-relaxing substrates prolong trapping events, promoting sub-diffusive behavior, while fast-relaxing substrates reduce trapping and enable larger migration steps, facilitating super-diffusion. The model also captures the interplay between glassy adhesion dynamics, viscoelastic substrate relaxation, and motor-driven forces, demonstrating their collective role in shaping emergent migration behaviors. Importantly, our framework makes testable predictions regarding migration transitions across diverse substrate conditions and the effects of inhibiting active processes like actin polymerization and actomyosin contractility on the transitions. These predictions were validated through experiments on HT-1080 cells migrating on alginate and rBM-based hydrogels,

confirming the impact of substrate viscoelasticity on migration phenotypes.

Our findings are significant in the context of emerging recognition that many soft tissues exhibit substantial viscoelasticity[6]. Cells exert dynamic forces and induce deformations on substrates, eliciting a complex, time-dependent mechanical response that can significantly alter the substrate's original structure and features to which cells are sensitive. Biological tissues typically exhibit viscoelastic properties, characterized by both storage (elastic) and loss (viscous) moduli. For instance, brain tissue, known for its soft and highly dissipative nature, relaxes stress within seconds, whereas liver, skin, and muscle tissues exhibit relaxation times spanning tens to hundreds of seconds. Specifically, liver tissue has a stress relaxation half-time of ~50 s, while skin and muscle tissues display significantly longer timescales ~650 s. Notably, pathological changes alter tissue viscoelasticity, as seen in human breast cancer tissue, which relaxes stress within ~10 s—a marked shift from healthy tissue[29,30]. Our model findings suggest that these variations in stress relaxation dynamics profoundly influence cell migration behaviors within these pathological contexts, dictating the modes of cell migration. These insights underscore the critical role of viscoelastic properties in regulating cellular responses and highlight the potential for targeting tissue mechanics in therapeutic strategies.

To unravel the mechanosensing processes modulating cell migration modes in response to changes in ECM viscosity, we implemented a glassy motor-clutch model of force transmission. A key innovation in our model is the inclusion of a power-law distribution of off-rate time constants $\tau_{off}$ to capture the glassiness of adhesion dynamics. This approach allowed us to account for the non-uniformity in cell movement, which is crucial for explaining the intermittency observed in migration steps and trap times. The power-law distribution of $\tau_{off}$ provides a realistic representation of the complex and variable nature of protein unfolding and unbinding in focal adhesion clusters, enabling our model to predict both sub-diffusive and super-diffusive behaviors based on the non-uniform movement patterns of cells. Our model provided a framework for timescale analysis revealing that substrate viscoelasticity regulates cell migration by modulating adhesion dynamics through the interplay of three key timescales: the clutch dissociation timescale $\tau_{off}$, motor-stall timescale $\tau_l$, and substrate relaxation timescale $\tau_s$. The clutch timescale $\tau_{off}$ arises due to the glassy nature of protein unfolding in focal adhesion clusters, which we capture through a long-tailed distribution ranging from ~1 to 100 s. The dissociation timescale ranges represent the wide range of dissociation kinetics shown by adhesion cluster proteins in FRAP (fluorescence recovery after photobleaching) experiments[31]. The motor-stall timescale $\tau_l$ is influenced by the interaction between myosin motors and substrate stiffness, ranging from ~10 to 30 s. By tuning viscosity of the substrate, we adjusted the substrate relaxation timescale $\tau_s$ over a wide range from $10^{-2}$ to $10^3$ s. Our findings indicate that when the substrate relaxation timescale is higher than the motor timescale ($\tau_s > \tau_l$), cells exhibit sub-diffusive migration. Conversely, when the substrate relaxation timescale is lower than the motor timescale ($\tau_s < \tau_l$), cells demonstrate super-diffusive migration. This interplay among the timescales governs adhesion dynamics and consequently cell migration behavior.

Investigations of traction force generating machinery in cell migration dynamics reveals that inhibiting actin polymerization and non-muscle myosin II through small molecule intervention results in reduced cell migration diffusivity on fast-relaxing substrates. This effectively triggered a shift from super-diffusive to sub-diffusive motion, which was concurrently predicted by our model. The reduction in the number of myosin motors results in a decreased motor-stall timescale $\tau_l$, which shifts the migration diffusivity regime. The consistency between experimental outcomes and our simulations highlights the crucial role of actin/myosin activity in determining migration patterns within the framework of the glassy motor-clutch model.

 

Our findings establish substrate stress relaxation as a key regulator of cell migration by modulating cell step size and trap times. Given that metastasis is responsible for up to 90% of cancer-related deaths, understanding the mechanical cues that drive migration is crucial for developing targeted therapies[32]. Our in vitro results demonstrate that slower-relaxing substrates suppress persistent migration, suggesting that tuning tissue viscoelasticity could provide a strategy to limit metastatic progression. Future studies should integrate biochemical signaling with mechanical modeling to further elucidate how contractility, substrate plasticity, and tissue remodeling influence migration. These insights have broad implications for tissue engineering, wound healing, and metastasis, offering new directions for therapeutic interventions that harness the interplay between cell mechanics and microenvironmental properties.

## Methods

### Formulation of a glassy motor clutch model for cell migration

We developed a computational model based on the motor-clutch theory[3,10,33,34] that employs a 2D framework, featuring independent motor-clutch modules along the X and Y axes (Fig. 2A). In essence, each module consists of molecular clutches representing adhesion clusters and link F-actin filaments to the substrate. The clutches get stretched due to the activity of actomyosin contractility and exhibit random engagement and disengagement with association and dissociation rates $r_{on}$ and $r_{off}$, respectively. The binding of these clutches to actin filaments opposes retrograde flow driven by myosin motors, facilitating polymerization at the protruding end and propelling cell migration. The dynamics of molecular clutches are governed by the master equation:

$$\frac{dP_{b,i}}{dt} = (1 - P_{b,i})r_{on,i} - P_{b,i}r_{off,i} \tag{1}$$

Here, $P_{b,i}$ denotes the binding state of the $i^{th}$ clutch, determined using a Monte Carlo sampling approach. The force exerted by a bound clutch is defined as $F_{c,i} = k_c(x_{c,i} - x_s)$, where $k_c$ represents the molecular clutch stiffness, $x_{c,i}$ denotes the displacement caused by the sliding actin filament, and $x_s$ corresponds to the substrate displacement, which is dependent on its viscoelastic properties. The cumulative force transmitted to the substrate is expressed as the sum of the forces from all the clutches: $F_s = \sum k_c(x_{c,i} - x_s)$.

The extension and dynamics of molecular springs facilitate the transmission of forces between the cell and the substrate, enabling mechanical sensing of the viscoelastic properties of the substrate. We model the substrate as a standard linear solid (SLS) viscoelastic material according to the following equation:

$$(k_a + k_l)\eta \frac{dx_s}{dt} + k_a k_l x_s = k_a F_s + \eta \frac{dF_s}{dt} \tag{2}$$

In this equation, one branch of the SLS model comprises a spring of stiffness $k_l$, which dictates the long-term behavior of the substrate. The other branch consists of two elements in series: a dashpot with viscosity $\eta$ and a spring with stiffness $k_a$. Upon force application, the initial response is dictated by the deformation of the two springs with stiffness $k_a$ and $k_l$, representing instantaneous elastic behavior. However, the dashpot viscosity $\eta$ resists deformation over time until the stiffness relaxes to the long-term value $k_l$.

The relaxation properties of the substrate regulate the transmitted clutch force $F_c$, thereby influencing the clutch dissociation rate $r_{off}$. To model the force-dependent dissociation rate, we employ Bell's

model, where $r_{off}$ is expressed as:

$$r_{off} = \frac{1}{\tau_{off}} * \exp\left(\frac{F_c}{F_{bond}}\right) \tag{3}$$

In the context of molecular clutches, $F_{bond}$ denotes the characteristic rupture force and the dissociation time constant, $\tau_{off}$, represents the characteristic timescale at which the clutches disengage. The disengagement of clutches represents the dissociation of the adhesion cluster which consists of unfolding of proteins and subsequent unbinding of protein-protein bonds (detailed further in SI). In conventional motor clutch models, $\tau_{off}$ for adhesion proteins has been considered constant[35]. However, adhesion complexes consist of multiple proteins, such as integrin, talin, vinculin, and zyxin, each exhibiting distinct and variable folding and unfolding dynamics[19–21]. Moreover, the conformational energy landscape for protein unfolding is inherently rugged, containing numerous local minima separated by energy barriers[36]. In such a landscape, some unfolding pathways involve low energy barriers, allowing for rapid transitions, while others encounter high energy barriers, resulting in rare but prolonged unfolding timescales (Fig. 2B). These differences stem from several factors, including the positional flexibility of the protein domains, as well as the stochastic nature of protein-protein interactions within the complex. For instance, talin, a key adhesion protein, displays a highly stochastic unfolding response, manifesting multiple unfolded conformations for the same level of the applied force[21]. It has additionally been observed that $\tau_{off}$ for certain adhesion proteins, such as zyxin, depends on mechanical fluctuations rather than taking on a fixed, deterministic value[20]. These observations underscore the non-uniform and stochastic nature of the unfolding and unbinding processes, where external forces and internal fluctuations modulate the overall dissociation behavior. Given these complexities, we propose that the dynamics of the adhesion cluster, consisting of multiple interacting proteins, cannot be adequately described by a constant dissociation time constant, $\tau_{off}$. Instead, these clusters exhibit glass-like dynamics, characterized by a broad distribution of stochastically emerging relaxation timescales having a range of $\tau_{off}$ values.

To capture the overall glassy behavior of the supramolecular adhesion complex, we adopted a distribution of $\tau_{off}$ assigned to molecular clutches (Fig. 2C). A protein's transition between conformational states can be expressed using Markov state modeling that reveals a long-tail distribution for the transition path timescales[37]. Long-tail distributions are particularly suited to representing rare, infrequent events which arise from conformational adaptability and redundancy in unfolding and refolding pathways. As the adhesion cluster stretches due to actomyosin-generated retrograde flow, multiple adhesion proteins undergo unfolding and eventual unbinding. The coupled unfolding and unbinding process are referred to as dissociation in this study and are detailed in the Supplementary Note 2. To account for the heterogeneous transition pathways of both frequent and infrequent events, we adopt a power law distribution of $\tau_{off}$, expressed as $p(\tau_{off}) = \frac{|(\beta - 1)|}{\tau_{min}}\left(\frac{\tau_{off}}{\tau_{min}}\right)^{-\beta}$, where $\beta$ and $\tau_{min}$ represent the glass coefficient and characteristic dissociation time constant, respectively[22,23]. This function exhibits a gradual decay from a peak at $\tau_{min}$ for small values of $\beta$, to a steeper decay for large values of $\beta$, asymptotically approaching a delta function, thus recovering the conventional model with a single constant time coefficient (Fig. 2C, D). By employing this power law distribution for $\tau_{off}$, we can tune the molecular clutch dynamics, effectively capturing the glassy nature of adhesion protein mechanics.

Crucially, when $\beta \leq 3$, the power law distribution exhibits infinite variance (derived in the SI), thereby violating the central theorem. This leads to a non-Gaussian long-tailed distribution, distinguishing it from distributions with $\beta > 3$ that converge to a Gaussian form upon averaging. Such long-tailed behavior is characteristic of glassy dynamics[38],

a defining feature of complex systems, which include adhesion clusters, where relaxation timescales may span a wide range. The presence of these long tails implies that rare, extreme events significantly influence the system's behavior, underscoring the need for models that account for this inherent complexity in protein dissociation dynamics. Conventional motor-clutch modules with finite-variance off-rate distribution can be reduced to deterministic modules through mean-field theory[3,33], but the glassy model lacks a finite mean off-rate and cannot be reduced to deterministic behavior. This fundamental non-Gaussian character allows the glassy adhesion dynamics to enable the anomalous diffusivity, as shown in SI note 1, and the differences in migration trajectory statistics are shown in Supplementary Figs. 6, 7.

The probability of engaged clutches $P_b$, and disengaged clutches $1 - P_b$, can be derived by solving master Eq. (1) with the on-rate $r_{on}$ held constant and the off-rate $r_{off}$ expressed as $r_{off} = \frac{1}{\tau_{off}} e^{\frac{F_c}{F_b}}$, where $\tau_{off}$ is as described earlier. Engaged clutches have their top end bound to actin filaments and move with the retrograde flow velocity $V_r$. Disengaged clutches move with the substrate velocity, ensuring that clutch association occurs at zero extension[3,33]. Therefore, the displacement of each clutch can be expressed as:

$$\frac{dx_{c,i}}{dt} = P_{b,i} V_r + (1 - P_{b,i}) \frac{dx_s}{dt} \qquad (4)$$

The retrograde flow velocity for each clutch module is updated according to Hill's relation expressed as $V_r = V_u(1 - \frac{F_m}{F_{stall}})$, where the myosin motor force $F_m$ is determined based on the force equilibrium at the cell center (see SI). The migration velocity $V_m$ in each direction is calculated as the difference in retrograde flow between two protruding ends (Fig. 3A), given by $D(t + \delta t) = D(t) + V_m(t)\delta t$. By solving the aforementioned system of equations for the motor clutch modules, we can simulate cell migration trajectories and calculate the MSD as

$$MSD = \frac{1}{T} \sum \left(D(t) - D_o\right)^2 \qquad (5)$$

For Brownian motion, the MSD exhibits a linear dependence with time as $MSD = 2Kt$, where $K$ denotes the diffusion coefficient. Conversely, anomalous diffusion is characterized by a non-linear dependence, $MSD \propto t^\alpha$, with $\alpha$ representing the power-law coefficient used to classify the diffusion mode.

## Cell culture
Human fibrosarcoma cells HT-1080 (ATCC) were cultured using established protocols. HT-1080 cells were cultured in high-glucose Dulbecco's Modified Eagles Medium (Gibco) supplemented with 10% fetal bovine serum (FBS) (Hyclone), and 1% Penicillin/Streptomycin (Life Technologies). Cells were cultured in a standard humidified incubator at 37 °C in a 5% CO$_2$ atmosphere. Cells were maintained at sub-confluency and passaged every 2–3 days.

## Hydrogel preparation and mechanical Characterization
Interpenetrating (IPN) hydrogels comprised of alginate and reconstituted basement membrane (rBM) were prepared following previously established protocols[7]. Briefly, low molecular weight (<75 kDa) and high molecular weight (280 kDa) alginate was purchased from Provona UP VLVG NovaMatrix and FMC Biopolymer, Protanal LF 20/40 respectively. Alginate was dialyzed in deionized water for 4 days (MW cutoff of 3500 Da), treated with activated charcoal, sterile-filtered, lyophilized, and then reconstituted to 3.5 wt% in serum-free Dulbecco's modified Eagle's medium (DMEM) (Gibco). The alginate was crosslinked using calcium sulfate digydrate. LMW alginate was used to make fast relaxing substrate while HMW alginate was used to make slow relaxing substrate. The IPNs were formulated as 10 mg/ml alginate – 4.4 mg/ml rBM.

Mechanical characterization of IPNs was performed with a stress-controlled AR2000EX rheometer (TA instruments). IPNs were formulated using luer-lock syringes and directly deposited onto the Peltier plate. A 25 mm flat plate geometry was slowly lowered to make full contact with the gel and a plastic pipette was used to add mineral oil (Sigma) to the edges of the gel to prevent dehydration. During modulus measurement, a time sweep was performed at 1 rad/s, 37 °C, and 1% strain for 3 h to allow for the storage and loss moduli to equilibrate. For stress relaxation experiments, the time sweep was followed by applying a constant strain of 5% to the gel, at 37 °C, and the resulting stress was recorded over the course of 3 h.

## Microscopy
Live microscopy was performed using a laser scanning confocal microscope (Leica SP8) that is suitable for live imaging. Live imaging conditions were maintained at 37 °C and 5% CO$_2$. R18 membrane dye was reconstituted in DMSO to obtain a 10 mg/ml stock solution. HT-1080 cells were labeled with R18 dye (1:1000 dilution) and the labeled cells were tracked with a 10× NA = 0.4 air objective for 24 h. 60 µm stack images were acquired every 10 min and imaging parameters were adjusted to minimize photobleaching and avoid cell death.

## Inhibition studies
Pharmacological inhibitors were added to cell media 10 min before time-lapse microscopy experiments. The concentrations used for the inhibitors were 100 nM latrunculin A (Tocris Bioscience, actin polymerization inhibitor) and 25 µM ML-7 (Tocris, myosin light-chain kinase inhibitor)

## Statistics and reproducibility
Measurements were conducted using one to three independent biological replicates from separate experiments. All statistical analyses were carried out in GraphPad Prism (version 9.1.0 for Mac, GraphPad Software). In violin plots, dashed lines mark the median; in scatter plots, solid lines denote median values. Reported p-values have been adjusted for multiple comparisons when necessary. Further details on the statistical methods are provided in the corresponding figure captions. Immunofluorescence and live-actin imaging were each repeated in three independent experiments.

## Cell migration simulation based on the motor–clutch framework
We simulated cell traction and migration dynamics using a stochastic motor–clutch model implemented with a Monte-Carlo update scheme. At each time step, individual clutch binding/unbinding events and motor-driven displacements were sampled probabilistically according to their rate constants, enabling realistic fluctuations in force transmission. All simulations and downstream analyses, including trajectory extraction and force–velocity characterization, were performed in MATLAB (MathWorks)[39]

## Reporting summary
Further information on research design is available in the Nature Portfolio Reporting Summary linked to this article.

# Data availability
Raw time-lapse images used in this study have been deposited on Figshare[40] and are publicly available at (https://doi.org/10.6084/m9.figshare.30763754). The data generated in this study are available in the Source Data file provided with this paper. Source data are provided with this paper.

# Code availability
The codes used in this study to simulate glassy motor clutch modules and generate migration trajectories are deposited on Zenodo[41] and

publicly available at (https://doi.org/10.5281/zenodo.17781658). All the figures were created and assembled using Blender[42] and Inkscape[43].

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

## Acknowledgements

This work was supported by National Cancer Institute awards U54CA261694 (V.B.S.); National Institute of Biomedical Imaging and Bioengineering awards R01EB017753 (V.B.S.) and R01EB030876 (V.B.S.); National Institute of General Medical Sciences award R01GM155943 (V.B.S.); NSF Center for Engineering Mechanobiology Grant CMMI-154857 (V.B.S.); NSF Grant DMS-2347834 (V.B.S.); and NIH grant 1R01GM148535 (O.C.). We are deeply grateful to the reviewers for their thoughtful and constructive feedback, which significantly strengthened this work.

## Author contributions

V.S., Z.G., and V.B.S. conceived the project. V.S. led the development of the model and performed all computational and numerical analyses for

the glassy adhesion framework. K.A. and O.C. designed and conducted the experiments. All authors contributed to writing the manuscript.

## Competing interests

The authors declare no competing interests.

## Additional information

[1]Center for Engineering Mechanobiology, University of Pennsylvania, Philadelphia, PA, USA. [2]Department of Mechanical Engineering, University of Pennsylvania, Philadelphia, PA, USA. [3]Shu Chien-Gene Lay Department of Bioengineering, University of California, San Diego, La Jolla, CA, USA. [4]Program in Immunology, University of California, San Diego, La Jolla, CA, USA. [5]Moores Cancer Center, University of California, San Diego, La Jolla, CA, USA. [6]CAS Key Laboratory of Mechanical Behavior and Design of Materials, Department of Modern Mechanics, University of Science and Technology of China, Hefei, China. [7]Department of Mechanical Engineering, Stanford University, Stanford, CA, USA. [8]Chemistry, Engineering, and Medicine for Human Health (ChEM-H), Stanford University, Stanford, CA, USA. [9]Department of Materials Science and Engineering, University of Pennsylvania, Philadelphia, PA, USA. [10]These authors contributed equally: Vivek Sharma, Kolade Adebowale, Ze Gong. ✉e-mail: vshenoy@seas.upenn.edu

