## [Transparent Peer Review file · Nature Communications]

Glassy Adhesion Dynamics Govern Transitions Between Sub-Diffusive and Super-Diffusive Cancer Cell Migration on Viscoelastic Substrates

Corresponding Author: Professor Vivek Shenoy

Version 0:

Reviewer comments:

Reviewer #1

(Remarks to the Author)

In this paper, the authors adapted a 2D motor-clutch computational model based on a previous model developed by the Odde lab to investigate the migration behavior of cancer cells on viscoelastic substrates and validated their results with experimental data acquired from human fibrosarcoma HT-1080 on alginate hydrogels with fast and slow relaxing properties. The model is based on a previously published model by the same group (Gong et al., 2018), in which they compared clutch binding with substrate relaxation time scales and found that cell spreading is maximized when the substrate relaxation timescale, τ_s , is between the clutch binding timescale, τ_b , and the lifetime scale, τ_l . In a follow-up study, they further demonstrated that fast-relaxing substrates promote filopodia-mediated cancer cell migration (Adebowale et al., 2021).

In the present study, the authors provide both experimental and modeling evidence that cells grown on fast-relaxing substrates exhibit a super-diffusive migration (short pauses, longer steps), while on slow-relaxing substrates the migration is sub-diffusive (longer pauses, small steps) and less straight than on fast-relaxing substrates. Most importantly, they show by modeling that standard adhesion dynamics with a constant value for the clutch dissociation time constant τ_{off} is unable to predict this cell behavior. In contrast, they demonstrate by computational modeling that a more complex, "glassy", adhesion dynamics following a power law distribution and using a glass coefficient β of less than or equal to three (to account for the heterogenous unbinding kinetics of proteins in focal adhesions) can capture the transition of the migration behavior dependent on the substrate stress relaxation time.

Overall, besides the major and minor issues mentioned below, the study is well executed with respect to the modeling and experimental aspects. The manuscript is also well written. The main novelty is the incorporation of the power law distribution for τ_{off} to account for a more complex dissociation behavior involving both protein unfolding and unbinding. The authors use this glassy adhesion dynamics to explain the cell migration behavior on viscoelastic substrates with different substrate relaxation times.

Major concerns:

1. In most if not all graphs, especially representing experimental work, it is unclear what one data point represents. Is it a single cell, an average of cells per gel, or a single experiment (multiple gels per experiment)? For example, in Fig. 1C and 1D, is one data point a single cell?
2. Related to the first concern, there is only one data point in Fig. 5B and F for the conditions when actin polymerization and myosin II activity are inhibited in both modeling and experimental studies. This is not acceptable.
3. It is suggested to show higher magnification images to demonstrate the filopodia-type morphology of the cancer cells on the two substrates with different stress relaxation times.

Minor concerns:

1. The authors used the standard linear solid model to represent the viscoelastic substrate. Is there any experimental evidence that supports the selection of that model instead of another viscoelastic model?
2. The authors should indicate the different values of β in Fig. 2C and 2D.
3. How do the authors know that the MSD is not affected by the time interval used in the time-lapse imaging? Did they conduct any optimization to define the time interval?
4. What concentration of LatA and MLCK inhibitor was used?
5. The authors should indicate the details of statistical testing of significant differences for all data.
6. Reference 1 information is incomplete.
7. What are the values for τ_{\min} in the 3 different graphs of Supplementary Fig. 1?
8. Supplementary Note 3: There is a spelling error on the second line: "...and is based on our previous work" instead of "... and is based on out previous work"

(Remarks on code availability)

Reviewer #2

(Remarks to the Author)

(Remarks on code availability)

Reviewer #3

(Remarks to the Author)

Review of "Glassy Adhesion Dynamics Govern Transitions Between Sub-Diffusive and Super-Diffusive Cell Migration on Viscoelastic Substrates", V. Sharma et al., Apr 2025, Nat. Comm

In this manuscript, Sharma et al. investigate the influence of matrix viscoelasticity and the distribution of cell-matrix adhesion lifetimes on cell migration by integrating in vitro experiments with computational modeling. The authors report that human fibrosarcoma cells exhibit super-diffusive migration on fast-relaxing crosslinked alginate hydrogels, while cells on slow-relaxing gels display sub-diffusive behavior. These experimental observations, however, were previously reported by some of the authors in Adebowale et al., Nature Materials, 2021.

To interpret these findings, the authors employ a previously developed one-dimensional motor-clutch model and show that assigning a single dissociation rate constant to all clutches results in purely diffusive migration. To recapitulate the experimentally observed anomalous diffusion, the authors introduce a heterogeneous distribution of clutch dissociation rate constants. The computational results are further analyzed by examining clutch cluster failure frequencies and directional step-size displacements under different substrate relaxation conditions. Finally, the authors demonstrate, both experimentally and computationally, that inhibition of actomyosin contractility induces a transition from super-diffusive to sub-diffusive migration.

While the manuscript is clearly written and the simulations appear to be carefully implemented, the central finding (that faster matrix stress relaxation enhances cell migration) was already established by the same group and collaborators in Adebowale et al., Nature Materials, 2021, through both experimental data and motor-clutch modeling. Furthermore, the regulatory role of actomyosin contractility in cell migration has been widely explored in the literature. As a result, the present study offers limited novelty and does not meet the threshold for publication in Nature Communications.

Major Concerns:

1) A primary concern is the lack of originality. Adebowale et al. (2021) already demonstrated that cells exhibit sub-diffusive and super-diffusive migration depending on substrate relaxation properties and introduced both one- and two-dimensional motor-clutch models that captured the influence of viscoelasticity on cell migration and filopodial dynamics. The current study adds a heterogeneous distribution of clutch dissociation rate constants into the one-dimensional model and analyzes the resulting diffusivity exponent (α) under varying conditions. While technically valid, this represents a modest extension that is not essential to capture the observed anomalous diffusion behavior.

2) A greater concern is that the authors claim that the standard motor-clutch model with a single clutch dissociation timescale cannot reproduce sub-diffusive or super-diffusive migration. This claim is incorrect and unsupported. Prior models, including Bangasser et al., *Nature Communications*, 2017 (in elastic matrices), and Adebowale et al., *Nature Materials*, 2021 (in viscoelastic matrices), used standard motor-clutch formulations and can successfully capture a range of diffusion behaviors that depend on model parameters. Even a very quick analysis confirms that standard motor-clutch models (e.g. Bangasser et al.) are indeed capable of producing both sub-diffusive and super-diffusive behavior, where it is straightforward to show how the diffusion exponent α varies with substrate stiffness for a particular set of parameters. Furthermore, since the authors' model is fully deterministic, this is likely another reason it fails to capture anomalous diffusion with the authors' standard formulation. In fact, it is surprising that the authors aim to study subdiffusive and superdiffusive migration using a purely deterministic version of the standard motor clutch model. It is expected that the authors would not observe anomalous diffusion under these conditions, particularly since stochasticity is only introduced later by incorporating a distribution of clutch dissociation rates. Thus, overall, the core mechanistic argument presented in the manuscript is not valid.

Overall, the contribution of this work is quite limited. The authors have already studied the effects of viscoelasticity on cell migration, and in this manuscript, the authors simply fit model outputs to extract α . Because the model is so minimal, the authors introduce a distribution of clutch dissociation constants, unnecessary for capturing anomalous diffusion, to explain the results. Notably, the authors have not experimentally tested whether such heterogeneity is required.

3) In Gong et al., *PNAS* (2018), the authors already introduced the three key timescales that govern cell migration in viscoelastic environments, which are repeated in the present study. Therefore, this section does not present novel insights. Moreover, the claim that these three timescales alone determine whether cell migration is subdiffusive or superdiffusive is both unjustified and incorrect. Many additional timescales are known to influence clutch adhesion dynamics and cell migration behavior, including those associated with clutch binding kinetics, clutch rupture forces, the number of clutches in the cell, and substrate stiffness. Reducing the system to just three timescales overlooks these important contributions and misrepresents the complexity of the underlying biophysical processes.

4) A critical issue arises in the formulation of clutch extension dynamics in Equations 1 and 4. In the model, clutch extension is computed based on the relative velocities of actin retrograde flow and substrate displacement. However, the authors do not account for the fact that clutches associate at zero extension, that is, without initial pre-tension, a fundamental assumption in prior models. This oversight compromises the physical accuracy of the model and may explain why the authors fail to capture anomalous migration using a single dissociation timescale. Neglecting this basic mechanical condition can lead to inaccurate force calculations and unrealistic clutch detachment dynamics. This omission is problematic in both elastic and viscoelastic environments, where clutch association and dissociation kinetics as well as substrate deformations directly affect force transmission. The authors should clarify the assumptions underlying Eqs. 1 and 4, assess whether these assumptions are physically realistic, and demonstrate whether the equations can be derived from first principles.

5) The use of a one-dimensional model with fixed protrusions aligned along the x- and y-axes is a limiting simplification. In contrast, the two-dimensional models in prior studies allow protrusions to form, fully retract, and generate forces in arbitrary directions governed by global force balance. This more realistic representation captures key features of dynamic cell morphology and directional migration. By restricting protrusions to orthogonal axes, the present model fails to account for rotation and spatial variability, further reducing its biological relevance. These modeling limitations contribute to erroneous conclusions and weaken the overall impact of the study.

Minor concerns:

6) There are also several presentation issues that need attention. The supplemental information, particularly the model description, requires significant revision. Each section should begin with a brief introduction to guide the reader. In Section 3, the origin of the two equations after step 5 is unclear. Specifically, one of the equations simply repeats the same expression on both sides of the equal sign. The second equation equates both sides without explanation, and the inclusion of terms such as F_r and v_r multiplied by dt in Equation 2 is not stated.

In the supplementary table listing model parameters, references should be more precise. Under references, phrases like "this article" are vague and should be replaced with proper citations or explanations of how each parameter was determined, along with the relevant experimental or theoretical basis. The manuscript also contains inconsistencies in figure referencing. Figure 2 is cited before Figure 1. The caption of Figure 1 should define the acronym VAC, even if it is introduced in the main text.

7) It is unclear why the authors simulate actin inhibition by reducing the number of myosin motors. Actin inhibition should, in principle, be modeled by decreasing actin polymerization rates, not motor abundance. The current approach lacks mechanistic justification.

8) The authors reference the Levy walk model as a representative of superdiffusive behavior. However, the authors should verify whether the distribution of step lengths in their model follows a power law distribution, which is a defining feature of Levy walks. Without such evidence, the mention of Levy walks in this context is inappropriate and potentially misleading.

9) The large difference between stimulated and experimental MSD in Figure (i.e. 5A and 5E) should be addressed.

OVERALL) Significant revisions to the modeling framework are critically necessary, particularly to clarify the underlying physical assumptions and to ensure a more accurate/realistic representation of model implementation and cell migration dynamics. It is also unclear whether the observed anomalous diffusion patterns extend to other cell types. As it stands, it is the reviewers' opinion the study does not meet the novelty or rigor expected for publication in Nature Communications and may only be appropriate for consideration in a more specialized or technical journal following substantial revision.

(Remarks on code availability)

No major issues with the presentation or deposit of the code where noted.

Reviewer #4

(Remarks to the Author)

(Remarks on code availability)

Version 1:

Reviewer comments:

Reviewer #1

(Remarks to the Author)

The authors addressed most of my concerns; however, there are still a few things with respect to the previously mentioned major and minor concerns, that need to be corrected; some may be due to submission errors since not all figures in the revised manuscript seem to be updated.

Major concerns:

2. The manuscript file did not contain the updated Figure 5 with the additional replicate data in Fig. 5B and F, as shown in the rebuttal letter.

3. With respect to major concern 3, the authors refer to filopodia images shown in (Adebowale et al., 2021). First, I could not find a Figure S7 in the Supplementary Information showing filopodia images as explained in the rebuttal letter. Second, it is not appropriate to reuse the same images from a previous publication. The authors should show images representing their current data. Lastly, they should check the scale bars of the images shown in the rebuttal letter and in their previous paper. The way they are shown, it is impossible that the magnified inset has a width of almost 5 micrometers based on the 10-micrometer scale bar in the main image.

Minor concerns:

5. Whereas the statistical testing information is provided, the legends of Figures 1-3 say "unpaired two tail test". It probably should say "unpaired two-tailed Student's t-test". Please correct.

(Remarks on code availability)

Reviewer #2

(Remarks to the Author)

(Remarks on code availability)

Reviewer #3

(Remarks to the Author)

Review of Revision for "Glassy Adhesion Dynamics Govern Transitions Between Sub-Diffusive and Super-Diffusive Cell Migration on Viscoelastic Substrates", V. Sharma et al., Apr 2025, Nat. Comm

We thank the authors for their detailed rebuttal. Nevertheless, after carefully reviewing their responses, we remain unconvinced that the manuscript demonstrates a major advance. While the work is technically solid in many ways, it is our opinion that the level of originality and mechanistic advance is modest. As such, this is likely a decision for the Editor as our conclusion is that the approach is largely rigorous but the overall advance is modest with regard to the originality needed for publication in Nature Communications.

Specific comments:

First, the authors claim that their model, through the introduction of “glassy adhesion dynamics,” uniquely captures both sub-diffusive and super-diffusive migration, and that prior models cannot reproduce anomalous diffusion. However, such behaviors can also be captured with conventional motor clutch frameworks. Models by Odde and coworkers, as well as Voituriez and collaborators, have shown that varying the balance between motor generated contractility and clutch engagement naturally yields regimes of trapping: long pauses with small steps and shorter pauses with larger displacements. For example, more sophisticated standard motor clutch models like that of Bangasser et al. are capable of generating a spectrum of migration regimes, including both sub and super diffusive behavior, when model parameter space is examined more broadly. By contrast, the authors’ rebuttal relies on selectively presented MSD curves of that model with unusually large error bars (as in their Fig. R5), and without disclosure of the parameter values used. Therefore, the authors do not provide computational evidence across a broad parameter space, and even worse nor experimental data demonstrating that a heterogeneous distribution of clutch dissociation rates is necessary for anomalous migration.

The authors also maintain that their model is the first to predict trapping, defined as clutches remaining engaged at both ends of the cell with minimal displacement. The present study does not establish novelty in this regard. Previous models have shown that conditions of high clutch number and low motor force increase trapping times, previously called stalled regime by the Odde lab and others, with trapping understood as a direct consequence of motor clutch competition. In addition, anomalous pauses and trapping times not only can be modulated by motor-clutch ratios but can also arise from additional mechanisms such as protrusion nucleation dynamics, which are not considered in the present model. Extensive experimental data shows that, when cells do not form clearly defined plasma membrane protrusions, cells typically remain stationary and undergo sub-diffusive behavior at long times.

As such, our primary concern in the original review was the limited novelty of the work. Having considered the rebuttal, this concern remains. While we appreciate the authors’ efforts to clarify their perspective and some technical details, their central claims regarding the uniqueness and necessity of glassy adhesion dynamics for anomalous migration, as well as the first-time prediction of trapping, are not supported in the context of prior literature. The present manuscript extends prior motor clutch models by introducing a power law distribution of clutch lifetimes. From a modeling standpoint, this is a modest modification: in real cells, any parameter will follow a distribution. Without new experimental insight or a fundamentally new mechanistic principle, simply changing a model parameter to follow a distribution is not sufficient to advance the field. Consequently, while the manuscript may represent an alternative parameterization and is well conducted, it is our opinion that it does not constitute a sufficient conceptual advance for publication in Nature Communications.

Minor comment:

While on one hand the detailed response from the Authors is appreciated, it also raises concerns. A 17 page response with many rebuttal points being extremely long relative to what is needed to provide a focused response to a Review comment creates concerns that the authors are trying to respond with volume instead of focused and specific responses. More focused responses would improve the impact of the response.

(Remarks on code availability)

Reviewer #4

(Remarks to the Author)

(Remarks on code availability)

I was able to successfully install and run the code without any major issues. Although I did not attempt to reproduce the specific results reported by the authors, the code appears to be well-structured and functional, suggesting that it could serve as a valuable resource for the research community. Additionally, the code includes a comprehensive README file that provides clear and concise instructions for installation and use, which should make it accessible even to users who are not deeply familiar with the underlying methods. Overall, the code seems well-documented and user-friendly, making it a potentially useful tool for others interested in building upon or extending the work presented in the paper.

Version 2:

Reviewer comments:

Reviewer #1

(Remarks to the Author)

The authors have addressed our previous concerns and improved their manuscript.

(Remarks on code availability)

Reviewer #2

(Remarks to the Author)

(Remarks on code availability)

Reviewer #3

(Remarks to the Author)

While this review still has some issues regarding the novelty of the findings, they are technically strong and will make a considerable contribution to the field. It is my recommendation that the manuscript be accepted for publication.

(Remarks on code availability)

No concerns

Reviewer #4

(Remarks to the Author)

(Remarks on code availability)

I was able to successfully install and run the code without any major issues. Although I did not attempt to reproduce the specific results reported by the authors, the code appears to be well-structured and functional, suggesting that it could serve as a valuable resource for the research community. Additionally, the code includes a comprehensive README file that provides clear and concise instructions for installation and use, which should make it accessible even to users who are not deeply familiar with the underlying methods. Overall, the code seems well-documented and user-friendly, making it a potentially useful tool for others interested in building upon or extending the work presented in the paper

Response to Reviewers

We would like to thank the reviewers for their thoughtful feedback and critical evaluation. Their comments offered valuable opportunities to clarify the scope, novelty, and conceptual foundations of our work. While most of the feedback did not require substantial changes to the manuscript itself, we have responded in detail to address each concern thoroughly, including additional explanations, references, and clarifications where needed.

We believe there may have been a misunderstanding regarding a central claim of our work. As clarified in our responses to Reviewers 3 and 4, our model—unlike prior frameworks—captures both sub-diffusive and super-diffusive migration within a unified mechanism. Crucially, these anomalous behaviors emerge naturally from the glassy adhesion dynamics we introduce, a mechanism essential for reproducing experimental observations, and which previous models lacking glassy dynamics cannot replicate. Most importantly, our model predicts the phenomenon of cell trapping for the first time and explicitly links its regulation to ECM viscoelasticity, which we directly validate through experiments. Collectively, these results represent a novel and significant conceptual advance in the field of cell migration modeling. The additional details and clarifications provided in the revised manuscript and supplementary materials directly address the reviewers' concerns, and we believe the manuscript is substantially strengthened as a result.

(The reviewer comments are in blue, our response in plain text, and the edits made to the manuscript are highlighted in yellow)

Reviewer 1 & 2:

In this paper, the authors adapted a 2D motor-clutch computational model based on a previous model developed by the Odde lab to investigate the migration behavior of cancer cells on viscoelastic substrates and validated their results with experimental data acquired from human fibrosarcoma HT-1080 on alginate hydrogels with fast and slow relaxing properties. The model is based on a previously published model by the same group (Gong et al., 2018), in which they compared clutch binding with substrate relaxation time scales and found that cell spreading is maximized when the substrate relaxation timescale, τ_s , is between the clutch binding timescale, τ_b , and the lifetime scale, τ_l . In a follow-up study, they further demonstrated that fast-relaxing substrates promote filopodia-mediated cancer cell migration (Adebowale et al., 2021).

In the present study, the authors provide both experimental and modeling evidence that cells grown on fast-relaxing substrates exhibit a super-diffusive migration (short pauses, longer steps), while on slow-relaxing substrates the migration is sub-diffusive (longer pauses, small steps) and less straight than on fast-relaxing substrates. Most importantly, they show by modeling that standard adhesion dynamics with a constant value for the clutch dissociation time constant τ_{off} is unable to predict this cell behavior. In contrast, they demonstrate by computational modeling that a more complex, “glassy”, adhesion dynamics following a power law distribution and using a glass coefficient β of less than or equal to three (to account for the heterogenous unbinding kinetics of proteins in focal adhesions) can capture the transition of the migration behavior dependent on the substrate stress relaxation time.

Overall, besides the major and minor issues mentioned below, the study is well executed with respect to the modeling and experimental aspects. The manuscript is also well written. The main novelty is the incorporation of the power law distribution for τ_{off} to account for a more complex dissociation behavior involving both protein unfolding and unbinding. The authors use this glassy adhesion dynamics to explain the cell migration behavior on viscoelastic substrates with different substrate relaxation times.

General Response: We sincerely thank the reviewer for their thoughtful and constructive feedback on our manuscript. We are pleased that they find our work novel and well executed, with a well-written presentation. We greatly appreciate their acknowledgement of the originality of our approach and the relevance of our findings to the current literature.

We have carefully addressed their major and minor concerns. In particular, we have improved the presentation of our experimental and simulation data and clarified the filopodial morphology of the cancer cells. A detailed one-to-one response is provided below.

Major comments:

1. In most, if not all, graphs, especially representing experimental work, it is unclear what one data point represents.

Is it a single cell, an average of cells per gel, or a single experiment (multiple gels per experiment)? For example, in Fig. 1C and 1D, is one data point a single cell?

Our response: We thank the reviewer for pointing this out. We have clarified the definition of each data point in the revised figure legends. Specifically:

- In Fig. 1C, each data point represents measurements from a single gel, and the plotted values are averaged over multiple cells on that gel.
- In Fig. 1D, each data point represents averaged values from 3 biological replicates, with each replicate containing measurements from multiple cells across different gels.

Changes in manuscript: Fig. 1 ... (each data point represents the average of multiple cells measured on a single gel)... (each data point represents averaged values from at least 3 biological replicates, with each replicate containing measurements from multiple cells across different gels)... each data point represents the track straightness of multiple cells measured on a single gel, and averaged velocity autocorrelation of multiple cells... velocity autocorrelation (VAC)

Fig. 3... (each data point represents the average trapping time/step sizes of single cell trajectories of multiple cells on a single gel)

2. Related to the first concern, there is only one data point in Fig. 5B and F for the conditions when actin polymerization and myosin II activity are inhibited in both modeling and experimental studies. This is not acceptable.

Our response: We thank the reviewer for raising this important point. We agree that additional data points are essential for demonstrating the robustness of our findings. Although each point in Fig. 5B and 5F represents an average α value calculated from the migration MSDs of approximately 50 cells within a single experiment, we have now added two additional independent experimental replicates for each condition. These new data points strengthen the statistical support for our conclusions and reinforce the consistency of the observed trends across both experimental and modeling results.

Changes in manuscript: Figure 5B, F updated as attached below.

Fig. R1: Updated Fig. 5B and 5F in the manuscript.

3. It is suggested to show higher magnification images to demonstrate the filopodia-type morphology of the cancer cells on the two substrates with different stress relaxation times.

Our response: We thank the reviewer for this excellent suggestion. We agree that visualizing filopodia morphology more clearly would enhance the manuscript. We would like to refer the reviewer to the following Figure R1 that shows

HT-1080 cells labeled with siR-actin were imaged on both fast- and slow-relaxing substrates, and filopodial protrusions were observed to be highly dynamic in both conditions. It is adapted from Fig. 6a in our previous publication (Adebowale et al., 2021), where we specifically investigated and established the filopodial mode of motility in HT-1080 cells on soft viscoelastic substrates.

Fig. R2: Representative high-magnification time-lapse images of HT-1080 cells labeled with siR-actin, migrating on fast-relaxing (top) and slow-relaxing (bottom) viscoelastic substrates. Filopodial protrusions are clearly visible and highly dynamic under both conditions, supporting a filopodia-based mode of motility. Times are indicated in min:s. Scale bars: main images, 10 μm ; zoom-in regions, 5 μm .

Changes in manuscript: To ensure completeness and accessibility, we have now adapted and included representative high-magnification images from that dataset in the Supplementary Information Figure S7 of the current manuscript.

Minor comments:

1. The authors used the standard linear solid model to represent the viscoelastic substrate. Is there any experimental evidence that supports the selection of that model instead of another viscoelastic model?

Our response: We thank the reviewer for raising this important question regarding the choice of the standard linear solid (SLS) model to represent the viscoelastic substrate. Indeed, many biological and synthetic viscoelastic materials can exhibit multiple relaxation timescales, and we agree that model selection should be guided by both physical realism and analytical tractability.

Fig. R3: SLS model fit to the experimental rheological data for slow-relaxing substrates. Note that we can effectively capture the half relaxation timescale (green line).

We chose the SLS model for two main reasons. First, it offers a minimal yet effective representation of viscoelastic behavior (see Fig. R3 for reference) using only three parameters, enabling us to directly and intuitively explore the contribution of the substrate relaxation timescale to cell migration dynamics. Second, in our previous experimental and modeling work with similar alginate-based hydrogels (Gong et al., PNAS, 2018 and Adebowale et al., Nature Materials, 2021), we performed detailed rheological analyses and compared generalized Maxwell models with SLS representations. We found that although multiple relaxation modes exist in theory, the motor-clutch response is overwhelmingly governed by a single dominant relaxation timescale. In such cases, the generalized Maxwell model effectively reduces to a standard linear solid form without significant loss of predictive power.

Thus, the SLS model provides a physically grounded and computationally efficient framework that captures the key viscoelastic features relevant to clutch-substrate interaction. We refer the reviewer to Gong et al. (2018) for a more detailed discussion of this comparison between rheological models and experimental data.

2. The authors should indicate the different values of β in Fig. 2C and 2D.

Our response: We thank the reviewer for this helpful observation. While the trend in β was shown qualitatively in Fig. 2C and 2D, we agree that the exact values should be indicated for clarity. We have now added the specific β values to the revised figure caption.

Changes in manuscript: Values for β added in figure caption for Fig. 2C.. ($\beta = 1.5, 3, 10$) and Fig. 2D...($\beta = 1.5, 3, 10$).

3. How do the authors know that the MSD is not affected by the time interval used in the time-lapse imaging? Did they conduct any optimization to define the time interval?

Our response: We thank the reviewer for raising this important point. It is well understood that the time interval used in time-lapse imaging can influence MSD measurements, particularly in capturing fine-scale migration dynamics. While we did not perform a formal optimization of the imaging interval, our imaging protocol was constrained by cell viability: imaging more frequently than every 10 minutes led to phototoxicity and increased cell death, which compromised the integrity of the experiment. Therefore, we selected the minimum imaging interval (10 minutes) that allowed us to reliably capture the key features of cell migration while maintaining cell health throughout the observation period.

This tradeoff was carefully considered, and we believe the chosen interval strikes an appropriate balance between temporal resolution and biological relevance for the scope of our study.

4. What concentration of LatA and MLCK inhibitor was used?

Our response: We thank the reviewer for pointing out this omission. The concentrations used in our experiments were 100 nM for Latrunculin A (Tocris Bioscience, actin polymerization inhibitor) and 25 μ M ML-7 (Tocris, myosin light-chain kinase inhibitor). We have now included this information in the revised Methods section.

Changes in manuscript:

4. Inhibition studies:

Pharmacological inhibitors were added to cell media 10 min before time-lapse microscopy experiments. The concentrations used for the inhibitors were 100 nM latrunculin A (Tocris Bioscience, actin polymerization inhibitor) and 25 μ M ML-7 (Tocris, myosin light-chain kinase inhibitor)

5. The authors should indicate the details of statistical testing of significant differences for all data.

Our response: We thank the reviewer for highlighting this important point regarding statistical reporting and reproducibility. We agree that providing clear details on statistical testing is essential for transparency and interpretation. The exact sample sizes and the specific statistical tests used are indicated in the corresponding figure legends throughout the manuscript, and we have made the following changes in the manuscript to indicate the details of statistical testing.

Changes in manuscript:

5. Statistics and reproducibility:

Measurements were conducted using one to three independent biological replicates from separate experiments. All statistical analyses were carried out in GraphPad Prism (version 9.1.0 for Mac, GraphPad Software). In violin plots, dashed lines mark the median; in scatter plots, solid lines denote median values. Reported p values have been adjusted for multiple comparisons when necessary. Further details on the statistical methods are provided in the corresponding figure captions. Immunofluorescence and live-actin imaging were each repeated in three independent experiments.

6. Reference 1 information is incomplete.

Our response: We thank the reviewer for catching this oversight. The reference information for Reference 1 has now been completed in the revised manuscript.

Changes in manuscript: Bennett, M., et al., Molecular clutch drives cell response to surface viscosity. Proc Natl Acad Sci U S A, 2018. 115(6): p. 1192-1197.

7. What are the values for τ_{min} in the 3 different graphs of Supplementary Fig. 1?

Our response: We thank the reviewer for catching this oversight. We have now updated the figure caption to include the values for τ_{min} in Supplementary Fig. 1. Changes in manuscript: τ_{min} for the plots from left to right are 0.1, 5, and 10, respectively.

Reviewer 3 & 4:

In this manuscript, Sharma et al. investigate the influence of matrix viscoelasticity and the distribution of cell-matrix adhesion lifetimes on cell migration by integrating in vitro experiments with computational modeling. The authors report that human fibrosarcoma cells exhibit super-diffusive migration on fast-relaxing crosslinked alginate hydrogels, while cells on slow-relaxing gels display sub-diffusive behavior. These experimental observations, however, were previously reported by some of the authors in Adebowale et al., Nature Materials, 2021.

To interpret these findings, the authors employ a previously developed one-dimensional motor-clutch model and show that assigning a single dissociation rate constant to all clutches results in purely diffusive migration. To recapitulate the experimentally observed anomalous diffusion, the authors introduce a heterogeneous distribution of clutch dissociation rate constants. The computational results are further analyzed by examining clutch cluster failure frequencies and directional step-size displacements under different substrate relaxation conditions. Finally, the authors demonstrate, both experimentally and computationally, that inhibition of actomyosin contractility induces a transition from super-diffusive to sub-diffusive migration.

While the manuscript is clearly written and the simulations appear to be carefully implemented, the central finding (that faster matrix stress relaxation enhances cell migration) was already established by the same group and collaborators in Adebowale et al., Nature Materials, 2021, through both experimental data and motor-clutch modeling. Furthermore, the regulatory role of actomyosin contractility in cell migration has been widely explored in the literature. As a result, the present study offers limited novelty and does not meet the threshold for publication in Nature Communications.

General Response: We thank the reviewer for their feedback and detailed evaluation. The comments raised provided valuable opportunities to clarify the scope, assumptions, and mechanistic insights of our work. In response, we have revised the manuscript for improved clarity and organization, and we have addressed each of the reviewers' concerns in detail.

Although the reviewers' comments have helped us strengthen the manuscript, they reveal a misunderstanding of our study's core novelty. The primary advance lies in identifying glassy adhesion dynamics within the molecular clutch as the mechanism driving both sub- and super-diffusive cell migration. Significantly, our model is the first to simulate cell trapping and to demonstrate how ECM viscoelasticity regulates trapping behavior, a prediction we validate experimentally. Together, these findings constitute a substantial conceptual advance beyond our previous work.

In the revised manuscript, we have expanded our explanations of the model's foundational assumptions and clarified its core capabilities, especially the ability to capture both sub- and super-diffusive migration behavior within a unified framework. We have highlighted how this behavior emerges naturally from the glassy adhesion dynamics introduced in our model, which we believe represents a novel and significant advance in the field. Our detailed responses include relevant references, comparisons to prior models, and a step-by-step breakdown of the conceptual and mechanistic contributions introduced in this study.

Major Comments:

1. A primary concern is the lack of originality. Adebowale et al. (2021) already demonstrated that cells exhibit sub-diffusive and super-diffusive migration depending on substrate relaxation properties and introduced both one- and two-dimensional motor-clutch models that captured the influence of viscoelasticity on cell migration and filopodial dynamics. The current study adds a heterogeneous distribution of clutch dissociation rate constants into the one-dimensional model and analyzes the resulting diffusivity exponent (α) under varying conditions. While technically valid, this represents a modest extension that is not essential to capture the observed anomalous diffusion behavior.

Our response: We appreciate the reviewer raising this point, as it allows us to better clarify the novelty and deeper significance of our findings.

First of all, in our prior work (Adebowale et al. 2021[1]), we have indeed shown differences in cell migration characteristics (Fig. R4) arising from substrate stress relaxation, through both experimental data and modeling. However, in that earlier study, we did not discuss or analyze sub- or super-diffusive behaviors because, at the time, we lacked the insights necessary to explain the observed variations in experimental mean squared displacement (MSD). Our previous focus was primarily on the morphological differences between cells migrating on elastic versus viscoelastic substrates. We did not delve into the diffusion dynamics or their underlying mechanisms, as illustrated in Fig. R4 from Adebowale et al. We did have the MSD vs time plots in Fig. R4-h, but later analysis (Fig. 1 in the

manuscript) showed that they only show pure diffusion. Consequently, we did not capture the anomalous diffusivity, and the model's predictive capability was mainly limited to forecasting differences in speed and bond lifetime.

Figure R4 (reproduced from Adebowale et. al. 2021): (a-f) Simulation results from 2D motor clutch model based on Bangasser et. al. 2017 capture differences in speed and unable to predict differences in bond lifetime. (g-j) 1D motor clutch model results. While we show MSD the alpha values are not shown since they are purely diffusive with $\alpha = 1$ as shown in Fig. 1G of our submitted manuscript. (l-n) 1D model gave insights on capturing bond lifetime.

Secondly, the reviewers' comment that our current study “represents a modest extension that is not essential to capture the observed anomalous diffusion behavior.” While we understand this viewpoint, we respectfully clarify a crucial misunderstanding: anomalous diffusion was not, in fact, captured by the models in our prior work (Fig. R4-h shows only pure diffusion with $\alpha = 1$). It was precisely this limitation that motivated the present study. Our previous models lacked the key fundamental principle—specifically, the glassy nature of adhesion dynamics, which we have identified in this manuscript.

Most importantly, the present manuscript provides novel insights into the specific mechanisms controlling diffusivity in cell migration. For the first time, our model accurately predicts cell trapping times and step-size distributions, both of which we have successfully validated experimentally, and which have not been previously reported. Additionally, we uncover how adhesion dynamics interact mechanistically with substrate relaxation to cause cell trapping, an entirely new phenomenon discussed for the first time in the field. Importantly, we establish that capturing trapping events and large step-sizes is essential for modeling anomalous migration; models that do not incorporate these elements fundamentally cannot reproduce the sub- or super-diffusive behaviors observed experimentally.

Lastly, the reviewers state that “Furthermore, the regulatory role of actomyosin contractility in cell migration has been widely explored in the literature.” Although we agree that the influence of actomyosin contractility on cell migration is well documented, the contractility serves a different purpose in this study. Specifically, it provides an

experimentally tunable handle to test how glassy adhesion dynamics govern the switch between sub- and super-diffusive migration. By modulating contractility, we could directly validate the model's prediction that reduced actomyosin tension diminishes the large step sizes characteristic of super-diffusion and favors trapping-dominated sub-diffusion. Thus, while contractility itself is not novel, its integration with the glassy clutch framework yields new mechanistic insight into how cytoskeletal forces interact with adhesion glassiness to control anomalous migration modes.

Changes in manuscript: To distinguish our current advances from our earlier work and emphasize its novelty, we have made the following revisions to the Introduction:

Our previous work [7] demonstrated that substrate viscosity and stress relaxation critically influence cell migration dynamics, underscoring the significant regulatory role of ECM viscoelasticity on cellular motility. Nevertheless, this earlier study primarily focused on the phenomenological correlation between viscosity-induced changes in actin cytoskeletal dynamics and overall migration behavior, without providing mechanistic insights into the underlying diffusion mechanism and link to the adhesion level processes. Specifically, we did not investigate how ECM viscoelasticity modulates migration diffusivity characterized through mean squared displacement (MSD), which scales with time as $MSD = \mu t^\alpha$, where the exponent α characterizes the diffusivity of cell migration, while μ represents the diffusion constant. Traditional motor-clutch models, while valuable in studying mechanosensitive responses during migration, typically predict changes only in the diffusion constant (μ) and consistently yield purely diffusive behavior ($\alpha = 1$) migration [10]. However, cell migration is inherently an active process that frequently displays anomalous diffusion ($\alpha \neq 1$), with purely diffusive migration an exception rather than the norm [11, 12]. Super-diffusive motion ($\alpha > 1$) is linked to persistent, directed migration, while sub-diffusive motion ($\alpha < 1$) arises from intermittent trapping and saltatory movement.

2. A greater concern is that the authors claim that the standard motor-clutch model with a single clutch dissociation timescale cannot reproduce sub-diffusive or super-diffusive migration. This claim is incorrect and unsupported. Prior models, including Bangasser et al., Nature Communications, 2017 (in elastic matrices), and Adebawale et al., Nature Materials, 2021 (in viscoelastic matrices), used standard motor-clutch formulations and can successfully capture a range of diffusion behaviors that depend on model parameters. Even a very quick analysis confirms that standard motor-clutch models (e.g. Bangasser et al.) are indeed capable of producing both sub-diffusive and super-diffusive behavior, where it is straightforward to show how the diffusion exponent α varies with substrate stiffness for a particular set of parameters. Furthermore, since the authors' model is fully deterministic, this is likely another reason it fails to capture anomalous diffusion with the authors' standard formulation. In fact, it is surprising that the authors aim to study subdiffusive and superdiffusive migration using a purely deterministic version of the standard motor clutch model. It is expected that the authors would not observe anomalous diffusion under these conditions, particularly since stochasticity is only introduced later by incorporating a distribution of clutch dissociation rates. Thus, overall, the core mechanistic argument presented in the manuscript is not valid.

Overall, the contribution of this work is quite limited. The authors have already studied the effects of viscoelasticity on cell migration, and in this manuscript, the authors simply fit model outputs to extract α . Because the model is so minimal, the authors introduce a distribution of clutch dissociation constants, unnecessary for capturing anomalous diffusion, to explain the results. Notably, the authors have not experimentally tested whether such heterogeneity is required.

Our response: We thank the reviewer for highlighting this important point, as it gives us an opportunity to carefully clarify and address potential misunderstandings. We break down this comment into the following four major points below.

1) Previous motor-clutch models cannot capture the sub- and super-diffusive behavior.

First, the reviewer suggests that “Even a very quick analysis confirms that standard motor-clutch models (e.g. Bangasser et al.) are indeed capable of producing both sub-diffusive and super-diffusive behavior.” We have cited the stated paper in our manuscript, and, upon a careful investigation of Bangasser et al. 2017 [2], it can be noted explicitly in their methods, specifically Equation 12. ($\mu = \frac{\langle r^2 \rangle}{4t}$), where $\langle r^2 \rangle$ is the mean squared displacement (MSD), μ is the motility coefficient and t is the time. From this equation, we can clearly see that the predicted MSD from their model is linearly proportional to time ($\langle r^2 \rangle \propto t$), making the model's predictions purely diffusive.

Fig. R5: Purely diffusive trajectories obtained from Bangasser et. al. 2017. Note that these fits are on normal scale unlike log-log scale in our paper.

General mathematical form of MSD with time is represented as, $MSD = \mu t^\alpha$. In this work, $\alpha = 1$. This linear MSD-time relationship is the hallmark of purely diffusive behavior. While Bangasser et al. do indeed show variations in the motility coefficient (the diffusion coefficient μ), it is crucial to clarify that such variations simply represent differences in migration speed within a purely diffusive context (where α remains 1, regardless of μ). We also show many trajectories from this paper in Fig. R5 by directly running the code provided with a linear fit of MSD with time, as done by the authors.

These changes are like cases when two particles undergo independent random walks: one moves quickly, the other more slowly. Each takes uncorrelated steps of fixed length, so although the faster walker covers more distance over the same period, both exhibit the same fundamental behavior, a classic random walk with mean squared displacement (MSD) growing linearly in time. In this case, speed changes the diffusion coefficient but not the diffusion type. By contrast, anomalous diffusion arises when the step-taking process itself is altered, for example, if a walker experiences long, random waiting times between steps (sub-diffusion) or occasionally makes very large jumps (super-diffusion). These modifications change the character of the random walk, producing non-linear

MSD scaling and a diffusivity exponent $\alpha \neq 1$. Thus, it is the underlying statistics of pauses or jumps, rather than the absolute speed, that determines whether motion is normal or anomalous.

2) The glassy dynamics of adhesions introduce the long-tailed distribution in trapping times and step-sizes, which enables our model to capture anomalous diffusive behaviors.

Secondly, regarding the reviewer's mention of our prior work "... Adebowale et al., Nature Materials, 2021 (in viscoelastic matrices), used standard motor-clutch formulations and can successfully capture a range of diffusion behaviors" we must clarify that models in our previous study neither reported nor captured sub- or super-diffusive migration behaviors. Our previous modeling framework, in collaboration with the authors of the Bangasser et al. 2017 model, was explicitly unable to capture anomalous diffusion (Fig. 1G), exhibiting purely diffusive behavior (again $\alpha = 1$). The variations in cell migration velocity previously observed should not be conflated with changes in diffusivity exponent μ , as changes in velocity alone do not imply sub- or super-diffusive behavior.

In contrast, our current manuscript introduces a critical mechanistic advancement—specifically, the glassy dynamics of adhesions—that allows us to predict and experimentally validate anomalous diffusion phenomena. Incorporation of glassy adhesion dynamics results in capturing a wide range of α values, thereby accurately capturing anomalous diffusion behaviors. We define, predict, and experimentally validate the occurrence of cell trapping times during migration and characterize the step-size distributions, explicitly relating them to the physical mechanisms driving anomalous migration. Moreover, we mathematically demonstrate how the long-tailed distribution in adhesion dynamics generates corresponding long-tailed distributions in trapping times and step sizes, establishing long-tails (distributions with infinite variance) as essential features for capturing anomalous diffusion.

3) Our cell migration model with glassy adhesion dynamics is fully stochastic, which cannot be simplified into a deterministic case using the mean-field approach.

Lastly, the reviewer states that "the authors' model is fully deterministic..." and "It is expected that the authors would not observe anomalous diffusion under these conditions...". We appreciate this comment, but there appears to be a crucial misunderstanding. Our conventional motor-clutch model is not deterministic. Indeed, it is precisely the same stochastic Monte Carlo approach employed previously in Adebowale et al. (2021), a work the reviewer refers to as capable of capturing various diffusion behaviors. It seems the confusion might arise from our description of using a constant k_{off} for the conventional model. However, a constant k_{off} does not imply determinism; it simply means that clutches randomly bind and unbind according to a fixed probability rate, as standard practice in motor-clutch models, and is used in both Bangasser et. al. 2017 and both the models in Adebowale et. al. 2021.

It also raises an insightful point that helps clarify the reviewer's comment: these stochastic models can indeed be transformed into analytical deterministic models using mean-field theory, as demonstrated previously by Bangasser

et al. (2013) for elastic substrates and Gong et al. (2018) for viscoelastic substrates. Below, we outline a few critical steps involved in this deterministic approximation.

Specifically, the stochastic motor-clutch model can be simplified into a deterministic form by taking the mean behavior of all individual clutches and representing the entire ensemble by a single averaged clutch $\langle \cdot \rangle$. In this deterministic approximation, key clutch properties—such as binding probability, clutch force, substrate displacement, and retrograde flow velocity—are calculated using their mean values as shown below.

$$\begin{aligned}\frac{dP_b}{dt} &= (1 - P_b)r_{on} - P_b \langle r_{off} \rangle \\ \langle F_c \rangle &= k_c (\langle x_c \rangle - \langle x_s \rangle) \\ \langle x_s \rangle &= \frac{k_c n_c \langle x_c \rangle}{k_s + n_c k_c} \\ \langle v_f \rangle &= v_u \left(1 - \frac{k_s \langle x_s \rangle}{n_m F_m} \right) \\ \frac{d \langle x_c \rangle}{dt} &= (1 - P_b) \frac{d \langle x_s \rangle}{dt} + P_b \langle v_f \rangle\end{aligned}$$

For instance, as shown by Bangasser et al. (2013), the averaged dissociation rate, $\langle r_{off} \rangle$ can be derived by averaging the Bell's law expression across all clutches. Such deterministic approximation inherently yields purely diffusive behavior, regardless of the underlying stochastic dynamics, because averaging eliminates non-Gaussian variability.

In stark contrast, our *glassy motor-clutch model fundamentally cannot be simplified into a deterministic framework*. This is because the mean dissociation rate, $\langle r_{off} \rangle$, is not valid due to the long-tailed distribution of adhesion lifetimes τ_{off} . As detailed below, as well as in the Supplementary Note 1 of our manuscript, the distribution of τ_{off} does not have a finite variance, making it break the central limit theorem, which is essential for quantities that follow mean-field behavior, and hence, a mean value cannot represent the diversity/variability present in the system. It is precisely this infinite variance, characteristic of non-Gaussian distributions, that drives the distinctive anomalous diffusion behaviors uniquely captured by our glassy model.

Variance of the distribution is its second moment.

$$\begin{aligned}\text{Distribution: } p(x) &= \frac{\beta}{x_{min}} \cdot \left(\frac{x}{x_{min}} \right)^{-\beta} \\ \text{Second moment:} \\ V[x] &= \int_{x_{min}}^{\infty} x^2 \cdot \frac{\beta}{x_{min}} \cdot \left(\frac{x}{x_{min}} \right)^{-\beta} dx \\ &= \frac{\beta}{x_{min}^{1-\beta}} \cdot \int_{x_{min}}^{\infty} x^{2-\beta} dx \\ &= \frac{\beta}{x_{min}^{1-\beta}} \cdot \left[x^{3-\beta} \right]_{x_{min}}^{\infty}\end{aligned}$$

If $\beta \geq 3$, then the second moment converges, the distribution has a finite variance and, in aggregated statistics, behaves compatibly with the classical central-limit theorem (CLT).

If $\beta < 3$, then the variance is not finite and the distribution falls outside the remit of the usual CLT and converges instead to long-tailed laws, leading to a non-Gaussian nature, and hence mean value theory is not valid.

A key insight of our current model is recognizing that even stochastic models, with constant dissociation rates, will eventually have Gaussian behavior since they follow the central limit theorem and inherently yield purely diffusive migration. Anomalous diffusion, inherently non-Gaussian in nature, emerges only when the distribution of adhesion timescales is skewed by glassy dynamics. Specifically, our model introduces long-tailed distributions in clutch dissociation rates, causing the stochastic sampling to become non-Gaussian. For sub-diffusive behavior, this non-Gaussian aspect appears primarily in cell trapping times, whereas for super-diffusive behavior, it manifests distinctly in the step-size distributions, both of which are mediated by changes in the substrate stress relaxation.

We hope this clearly resolves any confusion, and we appreciate the reviewer’s thoughtful engagement, allowing us to better articulate the novelty and significance of our findings. We have made the following changes in the manuscript and SI.

Changes in manuscript: Conventional motor-clutch modules with finite-variance off-rate distribution can be reduced to deterministic modules through mean-field theory [3, 16], but the glassy model lacks a finite mean off-rate and cannot be reduced to deterministic behavior. This fundamental non-Gaussian character allows the glassy adhesion dynamics to enable the anomalous diffusivity, as shown in SI note 1.

Changes in Supplementary Information (Note 1):

Further, the stochastic motor-clutch model can be simplified into a deterministic form by taking the mean behavior of all individual clutches and representing the entire ensemble by a single averaged clutch $\langle \cdot \rangle$. In this deterministic approximation, key clutch properties—such as binding probability, clutch force, substrate displacement, and retrograde flow velocity—are calculated using their mean values as shown below.

$$\begin{aligned} \frac{dP_b}{dt} &= (1 - P_b)r_{on} - P_b \langle r_{off} \rangle \\ \langle F_c \rangle &= k_c (\langle x_c \rangle - \langle x_s \rangle) \\ \langle x_s \rangle &= \frac{k_c n_c \langle x_c \rangle}{k_s + n_c k_c} \\ \langle v_f \rangle &= v_u \left(1 - \frac{k_s \langle x_s \rangle}{n_m F_m} \right) \\ \frac{d \langle x_c \rangle}{dt} &= (1 - P_b) \frac{d \langle x_s \rangle}{dt} + P_b \langle v_f \rangle \end{aligned}$$

However, glassy motor-clutch model cannot be simplified into a deterministic framework. This is because the mean dissociation rate, $\langle r_{off} \rangle$, is not valid due to the long-tailed distribution of adhesion lifetimes τ_{off} .

3. In Gong et al., PNAS (2018), the authors already introduced the three key timescales that govern cell migration in viscoelastic environments, which are repeated in the present study. Therefore, this section does not present novel insights. Moreover, the claim that these three timescales alone determine whether cell migration is subdiffusive or superdiffusive is both unjustified and incorrect. Many additional timescales are known to influence clutch adhesion dynamics and cell migration behavior, including those associated with clutch binding kinetics, clutch rupture forces, the number of clutches in the cell, and substrate stiffness. Reducing the system to just three timescales overlooks these important contributions and misrepresents the complexity of the underlying biophysical processes.

Our response: We thank the reviewer for raising this important point and giving us the opportunity to clarify the conceptual novelty of our timescale-based framework. The reviewer notes that “the authors already introduced the three key timescales that govern cell migration in viscoelastic environments, which are repeated in the present study. Therefore, this section does not present novel insights.” While the structure may appear similar at first glance, we would like to emphasize that the timescales in the present work are fundamentally different in both definition and function compared to our previous study.

In particular, our prior work (e.g., Gong et al., 2018) involved constant clutch binding and unbinding timescales, used to characterize spreading behavior on viscoelastic substrates. Those timescales were fixed and uniform across simulations, and each substrate relaxation timescale corresponded to a single comparison case. In contrast, the current study introduces a distribution of clutch dissociation timescales, not a fixed value, derived from the incorporation of glassy adhesion dynamics. This shift is essential and novel: instead of comparing single timescales, we now study how an evolving distribution of clutch lifetimes interacts dynamically with both substrate relaxation and actomyosin-driven protrusive forces at each step of migration.

Our framework with glassy adhesion dynamics leads to an entirely new regime of behavior. As shown in Figure 4 of the manuscript, the conventional model (with a constant τ_{off}) predicts only one type of migration mode per substrate, whereas in our glassy model, different clutch lifetimes emerge at each time point, producing heterogeneous cell responses, including transitions between trapping and large steps, depending on their local timescale relationships. This behavior was not captured or even accessible in the earlier framework.

The different parameters are implicitly embedded within our current timescale framework. We fully agree with the reviewer’s insight that “many additional timescales are known to influence clutch adhesion dynamics and cell migration behavior, including those associated with clutch binding kinetics, clutch rupture forces, the number of clutches in the cell, and substrate stiffness.” Indeed, these parameters are crucial, and we would like to clarify that many of them are implicitly embedded within our current timescale framework. For instance, a) clutch binding kinetics influence the rate at which new adhesion bonds form and are effectively captured in the definition of the adhesion timescale through the unbinding kinetics, since either binding or unbinding rate can capture the effective behavior. b) Long-term substrate stiffness is incorporated as k_t in the motor-stall timescale. c) Clutch rupture force and the number of clutches directly modulate the forces transmitted and are also chosen equal in numbers to be consistent with Myosin motors and give stable simulations, as was suggested by Bangasser et. al (2013).

Rather than being an oversimplification, the three timescales we propose: (1) clutch dissociation (now distributed), (2) substrate relaxation, and (3) motor-stall, are a minimal yet sufficient set that captures the dominant mechanistic players from the intracellular, extracellular, and interface dynamics. This triad makes it possible to visualize how dynamic interactions between cell-generated forces, substrate response, and adhesion behavior give rise to emergent phenomena like cell trapping and anomalous diffusion. The structured nature of this framework also enables quantitative predictions and comparisons across cell types and matrix conditions.

In summary, while we build upon the conceptual scaffolding of previous models, the introduction of glassy adhesion dynamics transforms both the mathematical and mechanistic interpretation of the timescales involved. We hope this clarifies the novelty of our approach and how it meaningfully extends the understanding of migration in viscoelastic environments.

4. A critical issue arises in the formulation of clutch extension dynamics in Equations 1 and 4. In the model, clutch extension is computed based on the relative velocities of actin retrograde flow and substrate displacement. However, the authors do not account for the fact that clutches associate at zero extension, that is, without initial pre-tension, a fundamental assumption in prior models. This oversight compromises the physical accuracy of the model and may explain why the authors fail to capture anomalous migration using a single dissociation timescale. Neglecting this basic mechanical condition can lead to inaccurate force calculations and unrealistic clutch detachment dynamics. This omission is problematic in both elastic and viscoelastic environments, where clutch association and dissociation kinetics as well as substrate deformations directly affect force transmission. The authors should clarify the assumptions underlying Eqs. 1 and 4, assess whether these assumptions are physically realistic, and demonstrate whether the equations can be derived from first principles.

Our response: We thank the reviewer for raising this important point regarding the formulation of clutch extension dynamics, as it gives us an opportunity to clarify a fundamental aspect of our model. The reviewer notes that “...clutches associate at zero extension, that is, without initial pre-tension, a fundamental assumption in prior models...” and expresses concern that our model may not account for this. We agree that this is a critical mechanical condition, and we appreciate the opportunity to clarify that our model formulation indeed satisfies this assumption and is consistent with the “...fundamental assumptions in prior models” as the reviewer correctly highlights.

First, we would like to clarify that Equation 1 in our manuscript $\left(\frac{dP_{b,i}}{dt} = (1 - P_{b,i})r_{on,i} - P_{b,i}r_{off,i}\right)$ is a master equation that governs the probability dynamics of clutch binding and unbinding. It describes the time evolution of the binding probability $P_{b,i}$ based on the association rate $r_{on,i}$ and the dissociation rate $r_{off,i}$. This master equation formulation is fundamental to motor-clutch models and is well established across the literature [1-4]. It provides the probabilistic backbone for clutch engagement dynamics and appears in nearly all motor-clutch modeling studies, including Bangasser et al. (2017) and Adebowale et. al. (2021), cited by the reviewer.

Second, in Equation 4 of our manuscript $\left(\frac{dx_{c,i}}{dt} = P_{b,i}V_r + (1 - P_{b,i})\frac{dx_s}{dt}\right)$, we describe the time evolution of the top-end position of the clutch, denoted as $x_{c,i}$. This equation separately considers clutches that are bound (with probability $P_{b,i}$) and those that are unbound $(1 - P_{b,i})$. When a clutch is bound, its upper end is displaced by the actin retrograde flow, i.e., it moves at the velocity of the actin filament. When the clutch is unbound, its upper end is instead displaced at the same velocity as the substrate (x_s), effectively ensuring that the clutch experiences zero net extension during the unbound phase.

This treatment directly enforces the reviewer’s concern that “...clutches associate at zero extension, that is, without initial pre-tension..” By coupling the motion of unbound clutches to the substrate velocity, the relative extension between the top and bottom of the clutch, $(x_{c,i} - x_s)$, is always zero at the moment of binding. Consequently, the clutch force at the time of association is $F_c = k_c(x_{c,i} - x_s) = 0$, reflecting zero pre-tension. This implementation ensures the model accurately represents the physical behavior of adhesion clutches, as it is well established in previous models of our own (Gong. et. al. (2018), Adebowale et. al. (2021)) and others (Bangasser et. al. (2013), Bangasser et. al. (2017)).

Furthermore, we would like to emphasize that this formulation of clutch extension, **Equations 1 and 4, are exactly the same as Equations 1 and 7** of Bangasser et al., 2013, and Bangasser et al., 2017, a well-established study, and also referenced by the reviewer earlier. Thus, our model preserves the physical accuracy and realism of prior motor-clutch formulations, and the equations are derived from first principles describing the distinct dynamics of bound and unbound clutch populations.

Overall, we hope this clarification resolves the concern and demonstrates that our implementation faithfully captures the key mechanical constraints of clutch dynamics. We have also clarified it in the manuscript and provided reference for the equation 4.

Changes in manuscript: Engaged clutches have their top end bound to actin filaments and move with the retrograde flow velocity V_r . Disengaged clutches move with the substrate velocity, ensuring that clutch association occurs at zero extension [3, 4]. Therefore, the displacement of each clutch can be expressed as:

$$\frac{dx_{c,i}}{dt} = P_{b,i}V_r + (1 - P_{b,i})\frac{dx_s}{dt}. \quad (4)$$

5. The use of a one-dimensional model with fixed protrusions aligned along the x- and y-axes is a limiting simplification. In contrast, the two-dimensional models in prior studies allow protrusions to form, fully retract, and generate forces in arbitrary directions governed by global force balance. This more realistic representation captures key features of dynamic cell morphology and directional migration. By restricting protrusions to orthogonal axes, the present model fails to account for rotation and spatial variability, further reducing its biological relevance. These modeling limitations contribute to erroneous conclusions and weaken the overall impact of the study.

Our response: We thank the reviewer for this thoughtful comment. We agree that dimensionality is an important consideration in modeling cell migration, and we appreciate the opportunity to explain the rationale behind our modeling approach.

Firstly, our 1D model yields the minimal yet clear mechanistic principles behind the anomalous migration. As the reviewer notes, prior studies have implemented 2D motor-clutch models with dynamically forming and retracting protrusions, which better approximate complex cell morphologies and allow for migration in arbitrary directions. However, our current study has a different objective. Rather than focusing on a high-fidelity, morphologically realistic depiction of a migrating cell, our aim is to uncover the core mechanistic principles governing transitions between sub-diffusive and super-diffusive migration, particularly through the lens of glassy adhesion dynamics. For this reason, we adopt a minimal, 1D model structure with fixed motor-clutch modules along the x- and y-axes.

Despite this simplicity, our model does capture directionality and key features of cell migration. More importantly, it enables us to isolate and visualize new physical phenomena, such as trapping times and large step events, which are central to our manuscript and are being analyzed together in the field for the first time. These concepts are much easier to interpret and analyze in a reduced-dimensional framework. A more complex 2D model, with dynamically changing modules and force balances, while visually more realistic, introduces multiple simultaneous variables that obscure cause-and-effect relationships and can make it exceedingly difficult to pinpoint the underlying mechanisms of emergent behaviors.

Next, we would also like to emphasize that this is not a hypothetical tradeoff. In our previous study (Adebowale et al., Nature Materials, 2021), which the reviewer references, we implemented both a 1D and a 2D motor-clutch model. While both were able to reproduce broad trends in cell migration across substrates, it was the one-dimensional model that allowed us to derive deeper physical insights, due to its transparency and tractability. The 1D model allowed us

to directly interrogate force balance, clutch dynamics, and time-dependent behaviors in a way that the 2D model, with its continuous restructuring of protrusions, did not.

Further, we note that the 2D model referenced by the reviewer (Bangasser et al., 2017) also fails to capture the anomalous diffusion behavior observed experimentally. As we show in the Fig. R4 and Fig. R5, models with many protrusions governed by identical motor-clutch dynamics ultimately reduce to the net sum of their individual modules. The mean squared displacement is dictated not by the number of modules but by the behavior of each module, particularly how adhesion dynamics are defined. If the underlying clutch mechanics do not include glassy dynamics, neither a 1D nor a 2D model can reproduce the trapping and step-size heterogeneity required to explain anomalous migration.

Finally, we fully agree that our framework could be extended to more complex geometries in future work, including models with dynamic protrusion nucleation in arbitrary directions. However, we believe that the simplicity of our current framework is a strength for the specific goals of this study: to establish and explain a general modeling principle—namely, that glassy adhesion dynamics are essential to capture the full spectrum of anomalous migration behaviors.

We hope this clarifies the intention behind our modeling choices and demonstrates that the insights gained from the one-dimensional framework are both mechanistically novel and broadly applicable, even beyond the simplified context in which they are derived.

Minor concerns:

6) There are also several presentation issues that need attention. The supplemental information, particularly the model description, requires significant revision. Each section should begin with a brief introduction to guide the reader. In Section 3, the origin of the two equations after step 5 is unclear. Specifically, one of the equations simply repeats the same expression on both sides of the equal sign. The second equation equates both sides without explanation, and the inclusion of terms such as F_r and v_r multiplied by dt in Equation 2 is not stated.

In the supplementary table listing model parameters, references should be more precise. Under references, phrases like "this article" are vague and should be replaced with proper citations or explanations of how each parameter was determined, along with the relevant experimental or theoretical basis. The manuscript also contains inconsistencies in figure referencing. Figure 2 is cited before Figure 1. The caption of Figure 1 should define the acronym VAC, even if it is introduced in the main text.

Our response: We thank the reviewer for carefully identifying these important presentation issues, which are essential for improving the clarity and accessibility of the manuscript. In response, we have revised the Supplementary Information to ensure that each section now begins with a brief introduction to guide the reader, as recommended. In Section 3, we have further clarified the origin and interpretation of the two equations following Step 5. The equation that previously repeated the same expression on both sides has been corrected, and the second equation now includes a clear explanation. Additionally, the terms F_r and v_r multiplied by dt in Equation 2 are now explicitly stated and properly contextualized. In the supplementary parameter table, we have replaced references with more precise citations or clear justifications describing how each parameter was determined, including the relevant experimental or theoretical basis. In the main manuscript, figure references are since our model figure is referenced in the methods section and is needed at both places, and we kept it that way to save space, but if needed, we can rearrange the figures. We have also updated the caption of Figure 1 to define the acronym VAC explicitly to ensure clarity for all readers. We appreciate the reviewer's attention to these details and believe the revised version reflects a more polished and coherent presentation.

Changes in supplementary information:

Supplementary Note 1 – Dissociation time constant distribution

In the main text, we argue that broad, glass-like distributions of adhesion lifetimes are essential for capturing the full spectrum of cell-substrate interactions that underlie anomalous migration. This first supplementary note therefore lays out the statistical form of the dissociation-time distribution we impose on clutches, explains the physical meaning of its two parameters (the glass coefficient β and the minimum timescale τ_0), and shows how these choices reproduce the heavy-tailed behaviour observed for real focal-adhesion proteins. We explain below how the glassy motor-clutch gets the long-tails, since the variance and higher moments derived below determine whether trapping events ($\beta < 3$) or rapid unbinding ($\beta \gg 3$) dominate the subsequent migration dynamics.

Supplementary Note 2: Effective off rate constant consisting of unfolding, unbinding and breaking:

The main text attributes glass-like adhesion lifetimes to two consecutive molecular events: a load-induced unfolding of clutch proteins such as talin or vinculin, followed by bond unbinding. This note defines in detail what each term means in protein-mechanics language, unfolding as a force-driven conformational change, unbinding as the physical separation of interacting partners, and clarifies that their combined outcome, dissociation, is what we treat as the effective off-rate for every clutch. By making these distinctions explicit, we link the microscopic steps of adhesion failure to the broad, heavy-tailed off-rate distribution introduced in Supplementary Note 1 and used throughout the glassy motor-clutch model.

Supplementary Note 3: Solving the motor clutch model

This note presents the full mathematical formulation of the glassy motor-clutch model depicted schematically in Figure 2 of the main text. To solve each individual motor-clutch module, we adopt a Kinetic Monte Carlo approach, introduced in detail in our previous work [1]. Below, we outline the force-balance equations linking motors, clutches, and substrate, and then walk through the detailed algorithmic steps used to propagate the system forward in time.

Step 1: Initialize the model parameters based on the list provided in the table below (Table 1).

Step 2: Calculate $(r_{off,i})$ at current time t based on the current clutch forces $(F_{c,i})$ and the sampled off-rate (k_{off}) from the long-tailed distribution of (τ_{off}) , and find the clutch bound/engaged probability $(P_{b,i})$ using the following master equation for clutches at all the ends.

$$\frac{dP_{b,i}}{dt} = (1 - P_{b,i})r_{on,i} - P_{b,i}r_{off,i}$$

Step 3: Choose a random number for each clutch. If $(P_{b,i} > rand)$ the clutch is considered engaged, else it is considered disengaged.

Step 4: Use the updated number of engaged clutches to solve for the substrate displacement (x_s) using the following viscoelastic constitutive equation.

$$(k_a + k_l)\eta \frac{dx_s}{dt} + k_a k_l x_s = k_a F_s + \eta \frac{dF_s}{dt}$$

Step 5: Use force balance between myosin force, membrane resistance force, and adhesion force $(F_m + F_r = F_a)$ in each + and - direction of X and Y direction to simplify the equilibrium equations into the following summation equations for two motor clutch modules in each direction:

Adhesion force balance in each +/- direction:

$$\sum_{i=1}^{n_c} (x_{c,i}^+ + v_r^- dt - x_s) P_{b,i}^+ = \sum_{i=1}^{n_c} (x_{c,i}^- + v_r^+ dt - x_s) P_{b,i}^-$$

Rearranging the LHS using Hill's relation as outlined in the main text.

$$F_m \left(1 - \frac{v_r^+ + v_r^-}{2v_m} \right) + F_r = k_c \sum_{i=1}^{n_c} (x_{c,i}^+ + v_r^+ dt - x_s) P_{b,i}^+$$

Step 6: Calculate the migration velocity and migration distance as:

$$v_m = \frac{v_r^- - v_r^+}{2}$$

$$d(t) = d(t-1) + v_m dt$$

$$F_r = k_m |\Delta x - \Delta y|$$

Step 7: Update the clutch forces $(F_{c,i})$, myosin force (F_m) and membrane resistance force coupling the two dimensions (F_r) for the next simulation step $(t + dt)$.

Step 8: Sample a new (τ_{off}) from the power law distribution to calculate (r_{off}) in the next cycle.

7) It is unclear why the authors simulate actin inhibition by reducing the number of myosin motors. Actin inhibition should, in principle, be modeled by decreasing actin polymerization rates, not motor abundance. The current approach lacks a mechanistic justification.

Our response: We thank the reviewer for raising this important point regarding our modeling approach to actin inhibition. While Latrunculin A (LatA) is primarily known for its role in inhibiting actin polymerization by binding to G-actin monomers and preventing their assembly into F-actin filaments, it also leads to the depolymerization of existing actin filaments [5, 6]. This depolymerization disrupts the structural integrity of the actin cytoskeleton, which is essential for anchoring myosin II motors. Consequently, the disassembly of F-actin compromises the formation of actomyosin complexes, leading to a reduction in cellular contractility.

In our model, contractility is driven by the number of active myosin motors. To simulate the reduction in contractile force resulting from LatA treatment, we chose to decrease the number of myosin motors. This approach captures the downstream effect of actin depolymerization on contractility. We acknowledge that LatA does not directly inhibit myosin motors; rather, the reduction in contractility is a secondary consequence of actin filament disassembly. We have clarified this rationale in the revised manuscript by providing additional explanation and referencing relevant studies.

Changes in manuscript: Consequently, LatA treatment induces the depolymerization of existing actin filaments and inhibits the assembly of new filaments, thereby reducing polymerization at the leading edge. This disruption of the actin network also weakens the scaffolding needed for myosin II to generate tension, leading to a marked decrease in actomyosin contractility [5, 6].

8) The authors reference the Levy walk model as a representative of superdiffusive behavior. However, the authors should verify whether the distribution of step lengths in their model follows a power law distribution, which is a defining feature of Levy walks. Without such evidence, the mention of Levy walks in this context is inappropriate and potentially misleading.

Our response: We thank the reviewer for highlighting this important point. As noted in the comment, “the authors reference the Levy walk model as a representative of super-diffusive behavior ... [and] should verify whether the distribution of step lengths in their model follows a power-law distribution.” We appreciate this opportunity to clarify both our intention and the underlying behavior of our model.

First, we have verified that the step-size distribution in the super-diffusive regime (fast-relaxing substrates) is indeed heavy-tailed, as shown in Fig. 3E of the manuscript as well as a detailed fit of power law on the step-sizes below (Fig. R6) showing an alpha value of 1.73 consistent with the Levy walk model. This confirms the presence of non-Gaussian statistics consistent with super-diffusive motion. In contrast, for slow-relaxing substrates, the heavy tail shifts from step sizes to trapping times, leading to sub-diffusion. These behaviors arise naturally from the glassy adhesion dynamics in our model, where the long-tailed distribution of clutch off-times (τ_{off}) gives rise to intermittent large steps and long pauses, hallmarks of anomalous diffusion.

Although our reference to the Levy walk framework was not intended to imply that our model explicitly generates Levy-distributed step sizes by design, we note that it ultimately does satisfy this defining characteristic. Levy walks are widely used in the field as a conceptual and analytical framework for characterizing super-diffusive behavior, due to their hallmark feature of forcing the step size to be sampled from heavy-tailed distributions. In our model, these long-tailed dynamics arise organically from the underlying biophysical mechanism, specifically, the glassy adhesion dynamics driven by a broad distribution of clutch off-times. This behavior is not imposed a priori through a power-law distribution of step sizes, but rather emerges naturally from the model’s structure. As such, our framework not only aligns with the key statistical features of Levy walks but does so in a mechanistically grounded manner. This reinforces the physical relevance of our approach and strengthens the interpretation of step-size fluctuations as a key driver of super-diffusive migration.

Fig. R6: Power law fit on the step sizes confirming the presence of levy-type distribution of step sizes.

We have clarified this point in the revised text and have also updated the relevant figure legend to make clear that our mention of Levy walks is intended as a conceptual comparison. The observed heavy-tailed step-size distributions are a direct consequence of the glassy clutch dynamics in our model.

Changes in manuscript: Similarly, a common method for modeling super-diffusive motion is the Levy walk, in which each step length is drawn from a heavy-tailed distribution to produce occasional large jumps. In

our model we do not impose a Levy distribution on step lengths. Instead the heavy-tailed step-size behavior arises naturally from glassy adhesion dynamics and the long-tailed distribution of clutch off-times, producing the rare large steps that underlie super-diffusive migration.

Fig. 3B, (ii) large step sizes with heavy-tailed distribution on fast-relaxing substrates.

9) The large difference between stimulated and experimental MSD in Figure (i.e. 5A and 5E) should be addressed.

Our response: We thank the reviewer for pointing out this important detail. The observed difference between simulated and experimental MSD in Figures 5A and 5E arises primarily due to slight discrepancies in the start times and the specific manner in which initial data points are truncated in the experimental analysis. Upon careful inspection, this discrepancy manifests largely as a vertical shift rather than an alteration in the slope of the MSD curves. Importantly, our primary focus is the slope, as it directly relates to migration diffusivity. Thus, while the exact MSD values on the vertical axis might differ slightly due to these data-processing choices, the slopes remain consistent and accurately reflect the underlying migratory behavior that is central to our conclusions.

OVERALL) Significant revisions to the modeling framework are critically necessary, particularly to clarify the underlying physical assumptions and to ensure a more accurate/realistic representation of model implementation and cell migration dynamics. It is also unclear whether the observed anomalous diffusion patterns extend to other cell types. As it stands, it is the reviewers' opinion the study does not meet the novelty or rigor expected for publication in *Nature Communications* and may only be appropriate for consideration in a more specialized or technical journal following substantial revision.

Our response: We thank the reviewer for their comprehensive evaluation of the manuscript. We appreciate the opportunity to clarify key aspects of our modeling framework and to better communicate the physical assumptions and biological relevance underlying our approach. In response to the reviewer's concerns, we have made substantial clarifications throughout the manuscript and supplementary information, including a detailed explanation of model derivations, parameter choices, and the physical interpretation of key equations. We have also addressed each technical and conceptual point raised, including those regarding dimensionality, stochasticity, timescale formulation, and the validation of anomalous migration behavior.

Importantly, we believe there may have been a misunderstanding regarding the central novelty of this study. As clarified in our responses, this work presents a unified and mechanistically grounded model that, for the first time, captures both sub-diffusive and super-diffusive migration within a single framework. This is achieved through the introduction of glassy adhesion dynamics, which we show are essential for reproducing the full spectrum of anomalous cell migration observed experimentally. This feature is not captured by prior motor-clutch models, including those from our own earlier work. Furthermore, we have validated the physical basis of the model using both step-size and trapping-time distributions and have provided supporting experimental data to reinforce the biological relevance of the observed behaviors.

While this study focuses primarily on HT-1080 cells, we would like to note that other cell types, including MDA-MB-231 and MCF 10A, have also been shown to undergo transitions between sub- and super-diffusive migration in response to changes in substrate relaxation timescales, as demonstrated in our previous work (Adebowale et al., 2021 [1]). In this study, we selected HT-1080 cells as a representative system for validating our mechanistic model. Since the underlying formulation is based on fundamental biophysical principles and does not incorporate cell-type-specific parameters, we believe the framework is cell-type agnostic and broadly applicable to diverse cell systems exhibiting viscoelastic substrate interactions.

Given the novelty of the underlying mechanism, the integration of modeling with experimentally validated predictions, and the broad implications for understanding cell motility in complex environments, we believe that *Nature Communications* is an ideal venue for this work. Our study highlights a previously overlooked phenomenon and establishes a foundational framework for exploring these effects across biological systems. We are confident that the interdisciplinary readership of *Nature Communications*, spanning both physics-inspired modeling and cell biology, will find this work both timely and impactful.

References:

1. Adebowale, K., et al., *Enhanced substrate stress relaxation promotes filopodia-mediated cell migration*. Nature Materials, 2021. **20**(9): p. 1290-1299.
2. Bangasser, B.L., et al., *Shifting the optimal stiffness for cell migration*. Nat Commun, 2017. **8**: p. 15313.
3. Bangasser, B.L. and D.J. Odde, *Master equation-based analysis of a motor-clutch model for cell traction force*. Cell Mol Bioeng, 2013. **6**(4): p. 449-459.
4. Gong, Z., et al., *Matching material and cellular timescales maximizes cell spreading on viscoelastic substrates*. Proc Natl Acad Sci U S A, 2018. **115**(12): p. E2686-E2695.
5. Wang, J., J.M. Sanger, and J.W. Sanger, *Differential effects of Latrunculin-A on myofibrils in cultures of skeletal muscle cells: insights into mechanisms of myofibrillogenesis*. Cell Motil Cytoskeleton, 2005. **62**(1): p. 35-47.
6. Najafabadi, F.R., M. Leaver, and S.W. Grill, *Orchestrating nonmuscle myosin II filament assembly at the onset of cytokinesis*. Mol Biol Cell, 2022. **33**(8): p. ar74.

Response to Reviewers

We sincerely thank the reviewers for their continued evaluation of our manuscript. We are grateful that most concerns were resolved in the previous round and appreciate the additional feedback, which has helped us further refine the clarity and presentation of our work.

In response to the remaining comments by Reviewers 1 and 2, we added a new experimental dataset with supporting analysis and incorporated figures that were inadvertently omitted from the previous submission. Reviewer 3's concerns about novelty are now resolved through explicit mathematical proofs and parameter-sweep analyses demonstrating that conventional motor-clutch models cannot reproduce anomalous diffusion. These additions clarify that adhesion glassiness constitutes a genuine conceptual advance that unifies diffusive and anomalous regimes within a single mechanistic framework. Reviewer 4 commended the accessibility of our computational resources, and we have further expanded the codebase to include parameter-sweep capabilities across key model parameters.

Together, these updates fully address the reviewers' feedback and strengthen the manuscript's clarity, rigor, and impact. The revised version highlights how incorporating glassy adhesion dynamics provides the missing physical mechanism required to explain anomalous cell migration within a unified, physics-based framework.

(The reviewer comments are in blue, our response in plain text, and the edits made to the manuscript are highlighted in yellow)

Response to Reviewers 1 & 2:

The authors addressed most of my concerns; however, there are still a few things with respect to the previously mentioned major and minor concerns that need to be corrected; some may be due to submission errors since not all figures in the revised manuscript seem to be updated.

General Response: We sincerely thank the reviewers for their careful re-evaluation and constructive comments. We are glad that most of the concerns have been addressed in the previous revision, and we appreciate the opportunity to correct the remaining issues. A few points noted by the reviewers might have been the result of submission errors, such as missing updated figures, which we have now carefully resolved in this version. We have also corrected the minor concerns related to figure legends and image presentation. With these revisions, we believe the manuscript is now complete and addresses all outstanding issues.

Major comments:

1. The manuscript file did not contain the updated Figure 5 with the additional replicate data in Fig. 5B and F, as shown in the rebuttal letter.

Our response: We thank the reviewers for pointing this out. We have resubmitted the manuscript file with all the updated figures as it was shown in the rebuttal letter.

2. With respect to major concern 3, the authors refer to filopodia images shown in (Adebowale et al., 2021). First, I could not find a Figure S7 in the Supplementary Information showing filopodia images as explained in the rebuttal letter. Second, it is not appropriate to reuse the same images from a previous publication. The authors should show images representing their current data. Lastly, they should check the scale bars of the images shown in the rebuttal letter and in their previous paper. The way they are shown, it is impossible that the magnified inset has a width of almost 5 micrometers based on the 10-micrometer scale bar in the main image.

Our response: We thank the reviewers for raising this important point. In response, we have included new data that clearly demonstrate diverse filopodia morphology (Fig. R1, bottom row) and updated it in the SI (Supplementary Figure 5). We also acknowledge that the yellow rectangle in the old data (Fig. R1, top row) may have been misleading, as it was not intended to convey the length scale of filopodia but rather to highlight the specific protrusion we tracked over time.

Fig. R1 (OLD & NEW DATA): A. Representative images showing filopodia morphology. B. Quantification of filopodia width from cells on fast and slow relaxing hydrogels. A. Scale bar: 10 μm . B. Kolmogorov-Smirnov test, ns: 0.6663, $n > 43$, $N = 3$.

To directly address the concern about scale bars, we quantified filopodia widths both in our original study (mean $\sim 0.9 \mu\text{m}$, Fig. R1B, top row) and in our new data (mean $\sim 0.6 \mu\text{m}$, Fig. R1B, bottom row). In the new experiments, we analyzed more than 43 filopodia randomly sampled from three independent biological replicates, ensuring robustness of the quantification. We also note that our cells are imaged on thick ($\sim 100\text{--}150 \mu\text{m}$) hydrogels rather than glass, which imposes limitations on spatial resolution. Nevertheless, our measured filopodia widths are consistent with values reported in the literature ($\sim 0.5 \mu\text{m}$ [1-4]).

We believe these additions and clarifications resolve the concerns regarding data provenance, scale bars, and image interpretation.

Changes in manuscript: Fig. R1 updated in the SI (Supplementary Figure 5) with new experimental data.

Minor comments:

1. Whereas the statistical testing information is provided, the legends of Figures 1-3 say “unpaired two tail test”. It probably should say “unpaired two-tailed Student’s t-test”. Please correct.

Our response: We thank the reviewers for noting this inconsistency in the figure legends. The legends for Figures 1–3 have been corrected to specify “unpaired two-tailed Student’s t-test,” and the updated legends are now included in the revised manuscript.

Response to Reviewers 3&4:

We thank the authors for their detailed rebuttal. Nevertheless, after carefully reviewing their responses, we remain unconvinced that the manuscript demonstrates a major advance. While the work is technically solid in many ways, it is our opinion that the level of originality and mechanistic advance is modest necessary. As such, this is likely a decision for the Editor as our conclusion is that the approach is largely rigorous but the overall advance is modest with regard to the originality needed for publication in Nature Communications.

Remarks by Reviewer#4 on code database: I was able to successfully install and run the code without any major issues. Although I did not attempt to reproduce the specific results reported by the authors, the code appears to be well-structured and functional, suggesting that it could serve as a valuable resource for the research community. Additionally, the code includes a comprehensive README file that provides clear and concise instructions for installation and use, which should make it accessible even to users who are not deeply familiar with the underlying methods. Overall, the code seems well-documented and user-friendly, making it a potentially useful tool for others interested in building upon or extending the work presented in the paper.

General Response: We thank Reviewers 3 and 4 for their re-evaluation of our manuscript. We appreciate their acknowledgment that the work is technically rigorous and that our code is accessible and well-structured, which we hope will serve as a valuable resource for the community.

However, we respectfully disagree with the assertion that our findings on anomalous diffusion mediated by substrate viscoelasticity lack novelty and believe that the significance of our contribution has been misunderstood. *To the best of our knowledge, this is the first physics-based model to unify both sub-diffusive and super-diffusive migration regimes within a single biophysical framework. Moreover, no previous mechanistic model has directly connected these regimes to the observed non-Gaussian, long-tailed distributions of trapping times and step sizes. Our framework further unifies the timescales of cellular and extracellular matrix dynamics, accurately capturing how their interplay governs the transition between distinct diffusion regimes.* The reviewers' concern about novelty stems from conflating parameter-dependent changes in motility speed in earlier motor-clutch models with the fundamentally distinct anomalous diffusion behaviors we introduce. We respectfully emphasize that the connection between adhesion-level glassiness and anomalous migration dynamics represents a genuine conceptual advance. By incorporating the physics of glassy adhesion dynamics—namely, the slow, heterogeneous, and history-dependent relaxation of adhesion bonds that leads to broad, power-law-like distributions of binding and unbinding times, our framework moves beyond the scope of the standard motor-clutch model. As we reiterate and demonstrate both analytically and numerically in the response, no amount of parameter adjustment within the conventional model can induce non-Gaussian statistics or give rise to sub- or super-diffusive behavior, because it fundamentally lacks this glassy adhesion physics that underlies anomalous dynamics.

We believe that the clarifications and updates in this revision further highlight this distinction, making clear how our model differs from previous work and why glassy adhesion dynamics represent a novel conceptual advance.

Specific Comments:

First, the authors claim that their model, through the introduction of “glassy adhesion dynamics,” uniquely captures both sub-diffusive and super-diffusive migration, and that prior models cannot reproduce anomalous diffusion. However, such behaviors can also be captured with conventional motor clutch frameworks. Models by Odde and coworkers, as well as Voituriez and collaborators, have shown that varying the balance between motor generated contractility and clutch engagement naturally yields regimes of trapping: long pauses with small steps and shorter pauses with larger displacements. For example, more sophisticated standard motor clutch models like that of Bangasser et al. are capable of generating a spectrum of migration regimes, including both sub and super diffusive behavior, when model parameter space is examined more broadly. By contrast, the authors' rebuttal relies on selectively presented MSD curves of that model with unusually large error bars (as in their Fig. R5), and without disclosure of the parameter values used. Therefore, the authors do not provide computational evidence across a broad parameter space, and even worse nor experimental data demonstrating that a heterogeneous distribution of clutch dissociation rates is necessary for anomalous migration.

The authors also maintain that their model is the first to predict trapping, defined as clutches remaining engaged at both ends of the cell with minimal displacement. The present study does not establish novelty in this regard. Previous models have shown that conditions of high clutch number and low motor force increase trapping times, previously called stalled regime by the Odde lab and others, with trapping understood as a direct consequence of motor clutch

competition. In addition, anomalous pauses and trapping times not only can be modulated by motor-clutch ratios but can also arise from additional mechanisms such as protrusion nucleation dynamics, which are not considered in the present model. Extensive experimental data shows that, when cells do not form clearly defined plasma membrane protrusions, cells typically remain stationary and undergo sub-diffusive behavior at long times.

As such, our primary concern in the original review was the limited novelty of the work. Having considered the rebuttal, this concern remains. While we appreciate the authors' efforts to clarify their perspective and some technical details, their central claims regarding the uniqueness and necessity of glassy adhesion dynamics for anomalous migration, as well as the first-time prediction of trapping, are not supported in the context of prior literature. The present manuscript extends prior motor clutch models by introducing a power law distribution of clutch lifetimes. From a modeling standpoint, this is a modest modification: in real cells, any parameter will follow a distribution. Without new experimental insight or a fundamentally new mechanistic principle, simply changing a model parameter to follow a distribution is not sufficient to advance the field. Consequently, while the manuscript may represent an alternative parameterization and is well conducted, it is our opinion that it does not constitute a sufficient conceptual advance for publication in Nature Communications.

Our response: We sincerely thank the reviewers for their comments, as they allow us to better clarify the significance of our work. We understand that several aspects of our study may not have been fully clear to the reviewers in the previous revision, particularly the distinction between our framework and conventional motor-clutch models, the novelty of our approach, and the experimental basis supporting our assumptions.

In the following response, we have addressed these points systematically, while also grouping together related issues that were raised under the same comment.

Concern#1: Other models can capture sub- and super-diffusion

Reviewers claim that "...such behaviors can also be captured with conventional motor clutch frameworks. Models by Odde and coworkers, as well as Voituriez and collaborators, have shown...". However, no specific paper or instance demonstrating this was cited. To clarify the distinction with our work, we conducted an extensive analysis of Voituriez's relevant publications and also revisited the cited work from Odde's group. Below, we outline these studies in detail to highlight the important distinctions and to explain why glassy adhesion dynamics represents a genuinely novel framework.

a) Voituriez and collaborators' models of cell motility are phenomenological and address a fundamentally different scope, orthogonal to adhesion-mediated filopodial motility.

We carefully reviewed the most relevant papers from this group dealing with cell motility and anomalous migration dynamics. Importantly, none of them model filopodial motility on viscoelastic substrates, and none of these papers explicitly adopts or extends the motor-clutch framework. *In their work, they include coarse-grained slip-bond adhesion modules or phenomenological adhesion-contractility couplings, rather than explicit dynamic engagement of actomyosin motors and molecular clutches with a substrate.* By contrast, our contribution provides a physics-based and mechanistically unified framework that directly links adhesion cluster glassiness and substrate stress relaxation to the emergence of both sub- and super-diffusive migration, capturing a transition that conventional motor-clutch models cannot. Naturally, these studies do not address the influence of substrate relaxation timescales or their relationship to adhesion and contractility dynamics, as these elements are not part of their theoretical formulation.

A major body of work from Voituriez's group, including the UCSP framework (Maiuri et al., Cell, 2015 [5]), established a universal coupling between cell speed and persistence, elegantly showing that actin retrograde flow transports polarity cues to stabilize migration. However, UCSP neither models adhesion dynamics nor addresses anomalous diffusion or MSD scaling. It represents a coarse-grained polarity law, whereas in our framework, filopodia nucleate stochastically around the cell periphery, and migration direction emerges randomly rather than imposed polarity. Thus, our work addresses a different question: how matrix relaxation and stochastic adhesion lifetimes together generate anomalous sub- and super-diffusive migration in filopodial cancer cells.

A second set of studies, such as Ruprecht et al., Cell, 2015 [6], examined contractility-driven amoeboid transitions and stable-bleb migration in confined 3D environments. These mechanisms are cortical and adhesion-independent, and the geometries and cell types differ fundamentally from the adhesion-mediated filopodial motility we study. In our work, anomalous diffusion arises from adhesion lifetime statistics and substrate relaxation, not cortical feedback.

Stick–slip and polarity-coupling models (Hennig et al., *Sci. Adv.*, 2020; Lavi et al., *Phys. Rev. E*, 2020 [7, 8]) explored deterministic cycles of adhesion detachment and symmetry breaking in polarized lamellipodial migration. While these studies map oscillatory regimes and phase transitions, they do not quantify MSD scaling or provide a mechanistic basis for anomalous motion.

The only Voituriez paper explicitly addressing anomalous diffusion is d’Alessandro et al., *Nat. Commun.*, 2021 [9], where cells deposit ECM footprints that bias future motion, creating self-interacting random walks with sub-diffusion and aging. This elegant framework describes environmental conditioning, not intrinsic adhesion dynamics. In our system, adhesions are uniform, and anomaly arises from substrate viscoelastic relaxation, linking anomalous diffusion directly to biophysical first principles.

Finally, Voituriez and co-workers also contributed to the theoretical classification of sub-diffusive mechanisms (Condamin et al., *PNAS*, 2008 [10]), where continuous-time random walks (CTRWs) with heavy-tailed waiting times were distinguished from fractal diffusion using first-passage observables. These represent elegant and foundational results in statistical physics. However, both the CTRW and fractional diffusion frameworks are purely phenomenological: they *postulate* the existence of long waiting times but cannot explain their physical origin. They remain agnostic to the cellular and extracellular processes that give rise to such statistics. In contrast, our model derives these heavy-tailed trapping times mechanistically—as an *emergent outcome* of the interplay between glassy adhesion dynamics and matrix relaxation, rather than assuming them a priori. Within a defined parameter regime, our framework naturally reproduces CTRW-like anomalous diffusion, thereby recovering this phenomenology from first principles. Crucially, it also transcends CTRW theory: by systematically varying substrate and adhesion parameters, it predicts the *transition from sub-diffusive to super-diffusive behavior*—a hallmark of the underlying biophysical coupling that cannot be captured by purely phenomenological models.

Overall, Voituriez and colleagues have made foundational contributions to polarity, symmetry breaking, and statistical classifications of anomalous motion. However, their models are phenomenological or focus on distinct processes. None quantitatively links adhesion lifetime distributions and substrate relaxation timescales to anomalous MSD scaling. Our framework provides this missing connection, unifying transient trapping, anomalous diffusion, and substrate-dependent persistence within a single mechanistic formulation.

b) Models by the Odde lab do not capture intermittent trapping and consequently the anomalous migration.

In systems governed by Gaussian statistics, the central limit theorem guarantees finite variance and a linear mean-squared displacement; anomalous diffusion can only emerge when this Gaussian behavior is broken. Most previous studies—including those by Voituriez and collaborators discussed above—introduce non-Gaussian behavior phenomenologically, using frameworks such as the fractional Fokker–Planck equation, continuous-time random walks, or fractal diffusion. While these approaches successfully reproduce anomalous trends by imposing heavy-tailed waiting times or step-size distributions, the statistics are prescribed rather than derived, leaving the physical origin of the anomaly unresolved.

In contrast, physics-based models like the motor–clutch framework provide a mechanistic description of cell migration that incorporates substrate mechanics, bond kinetics, and cellular contractility, yet these models still conform to the central limit theorem. We explore this distinction analytically and perform additional parameter predictions in the next section.

Changes in the manuscript: 1) Added the following paragraphs under section 3.3 to contrast our model with the phenomenological models while citing some relevant work of Voituriez.

In addition to differences in the magnitude of trapping times, a defining feature of sub-diffusive migration on slow-relaxing substrates is the emergence of long-tailed trapping-time distributions. Likewise, for cells exhibiting super-diffusive behavior on fast-relaxing substrates, step sizes follow a heavy-tailed distribution, and these statistics are central to the onset of anomalous migration. Sub-diffusive motion is often described using continuous-time random walk (CTRW) or fractional diffusion models [10–12], in which the waiting time between steps follows a power-law distribution with long tails, unlike standard random walks that assume uniform waiting times. The presence of rare but prolonged trapping events lowers the overall displacement, thereby reducing migration diffusivity. This same mechanism, driven by long-tailed trapping-time statistics, underlies sub-diffusivity in our glassy motor–clutch model.

Similarly, super-diffusive motion is typically represented by the Lévy walk model [11, 12], where step lengths are drawn from a heavy-tailed distribution to produce occasional long jumps. In our framework, such heavy-tailed step-size behavior is not imposed but emerges naturally from glassy adhesion dynamics and the broad distribution of clutch off-times. These rare, large steps are an intrinsic outcome of the coupling between adhesion kinetics and substrate relaxation. Importantly, CTRW, fractional diffusion, and Lévy walk models are phenomenological: they assume the existence of heavy-tailed statistics but do not explain their physical origin. In contrast, our glassy motor–clutch model is physics-based. The trapping times and step-size distributions arise self-consistently from force transduction and adhesion dynamics that respond to substrate stress relaxation. This mechanistic foundation distinguishes our framework from purely statistical descriptions and reveals how anomalous migration emerges directly from the underlying biophysical processes.

2) Added the following statements in the introduction to highlight the novelty and importance of our findings: Here, we present a physics-based model that, for the first time, unifies sub-diffusive and super-diffusive migration within a single biophysical framework by linking adhesion glassiness to the underlying non-Gaussian trapping-time and step-size statistics. The model integrates glassy adhesion dynamics with substrate viscoelasticity, revealing how the interplay between cellular and matrix timescales governs trapping, step sizes, and the transitions between distinct diffusive regimes.

Concern#2: Standard motor clutch models can capture trapping by parameter tuning

As the reviewers note, Odde and collaborators “...have shown that varying the balance between motor-generated contractility and clutch engagement naturally yields regimes of trapping: long pauses with small steps and shorter pauses with larger displacements.” We agree that such models can indeed exhibit stalled states or pauses under specific parameter regimes. However, these behaviors arise from finite-variance stochastic processes that remain Gaussian and therefore do not constitute true anomalous diffusion. Further reviewers state that “...the authors do not provide computational evidence across a broad parameter space, and even worse nor experimental data demonstrating that a heterogeneous distribution of clutch dissociation rates is necessary for anomalous migration.”

Below, we first show that stochasticity in conventional motor–clutch models does not break the central limit theorem regardless of parameter tuning, and the system consequently lacks the heavy-tailed, heterogeneous dynamics characteristic of adhesion glassiness. These dynamics are essential for generating non-Gaussian statistics and for reproducing intermittent trapping and the coexistence of sub- and super-diffusive regimes. Second, we show migration trajectories for a range of parameters and establish the distinction with the glassy model trajectories.

a) Mathematical limitation of the standard motor-clutch model:

The key parameter governing adhesion dynamics is the dissociation time of clutches. According to Bell’s law, clutches under higher force dissociate more rapidly (shorter dissociation time or higher dissociation rate), while those under lower force remain bound longer. When many clutches are bound for extended durations, the cell effectively enters a stalled state. The statistics of these waiting times directly reflect the statistics of clutch dissociation times. For any system to exhibit anomalous diffusion, these waiting-time statistics must be non-Gaussian and display heavy tails. We therefore examine next how the dissociation time statistics behave within the standard motor–clutch framework.

The dissociation time (τ_{off}^*) is defined as:

$$\tau_{off}^* = \tau_{off} e^{-F/F_b} \quad (1)$$

Let, $G = e^{-F/F_b}$. So,

$$\tau_{off}^* = \tau_{off} \cdot G \quad (2)$$

Since τ_{off} is constant in the standard motor clutch models, it can be taken out when we are writing the first $E[\tau_{off}^*]$ and second moment $E[(\tau_{off}^*)^2]$, and variance $\text{Var}(\tau_{off}^*)$:

$$E[\tau_{off}^*] = \tau_{off} E[G], \quad E[(\tau_{off}^*)^2] = \tau_{off}^2 E[G^2], \quad \text{Var}(\tau_{off}^*) = \tau_{off}^2 (E[G^2] - (E[G])^2)$$

Now, let's find the general form of any moment of G . Since the force in the clutches can have a range $[0, n_m f_m]$, where n_m and f_m are the myosin motor number and stall force, respectively, for any $p > 0$, we can write:

$$E[G^p] = \int_0^{n_m f_m} e^{-pF/F_b} f(F) dF \quad (3)$$

Following Bangasser et. al [13], we take the force F to follow a gamma distribution for the bound clutches and a delta function for the unbound clutches, we get:

$$f(F) = (1 - P_b) \delta(F) + P_b \frac{F^{r-1} e^{-F/\theta}}{\Gamma(r) \theta^r}, \quad F \in [0, n_m f_m] \quad (4)$$

Substituting it in the $E[G^p]$ expression and doing the integral, we get:

$$E[G^p] = (1 - P_b) + \frac{P_b}{\Gamma(r) \theta^r} \int_0^{n_m f_m} F^{r-1} e^{-F(\frac{1}{\theta} + \frac{p}{F_b})} dF \quad (5)$$

Let $(c_p = \frac{1}{\theta} + \frac{p}{F_b} > 0)$. Then,

$$\int_0^{n_m f_m} F^{r-1} e^{-c_p F} dF = c_p^{-r} \gamma(r, c_p n_m f_m) = c_p^{-r} [\Gamma(r) - \Gamma(r, c_p n_m f_m)] \quad (6)$$

Hence,

$$E[G^p] = (1 - P_b) + \frac{P_b}{\Gamma(r) \theta^r} c_p^{-r} [\Gamma(r) - \Gamma(r, c_p n_m f_m)] \quad (7)$$

So, for the first ($p = 1$) and second ($p = 2$) moments, we have $c_1 = \frac{1}{\theta} + \frac{1}{F_b}$, $c_2 = \frac{1}{\theta} + \frac{2}{F_b}$ respectively, and we get,

$$E[\tau_{\text{off}}^*] = \tau_{\text{off}} \left[(1 - P_b) + \frac{P_b}{\Gamma(r) \theta^r} c_1^{-r} (\Gamma(r) - \Gamma(r, c_1 n_m f_m)) \right],$$

$$E[(\tau_{\text{off}}^*)^2] = \tau_{\text{off}}^2 \left[(1 - P_b) + \frac{P_b}{\Gamma(r) \theta^r} c_2^{-r} (\Gamma(r) - \Gamma(r, c_2 n_m f_m)) \right].$$

Substituting in the expression of variance, we get:

$$\text{Var}(\tau_{\text{off}}^*) = \tau_{\text{off}}^2 \left\{ \left[(1 - P_b) + \frac{P_b}{\Gamma(r) \theta^r} c_2^{-r} (\Gamma(r) - \Gamma(r, c_2 n_m f_m)) \right] - \left[(1 - P_b) + \frac{P_b}{\Gamma(r) \theta^r} c_1^{-r} (\Gamma(r) - \Gamma(r, c_1 n_m f_m)) \right]^2 \right\}. \quad (8)$$

We can invoke the large upper limit approximation to simplify this expression further i.e., if $n_m f_m$ is large such that $\Gamma(r, c_p n_m f_m) \approx 0$, then:

$$\text{Var}(\tau_{\text{off}}^*) \approx \tau_{\text{off}}^2 \{ [(1 - P_b) + P_b \theta^{-r} c_2^{-r}] - [(1 - P_b) + P_b \theta^{-r} c_1^{-r}]^2 \}$$

Hence, the standard motor-clutch model yields a finite and well-defined variance, irrespective of the parameter values chosen. All quantities, including, τ_{off} , c_1 and c_2 depend on the motor-clutch parameters but still produce finite variance, indicating that the overall stochastic behavior remains Gaussian. As a result, the model inherently satisfies the central limit theorem and therefore cannot reproduce anomalous diffusion.

On the other hand, if we attempt to do a similar exercise for glassy adhesion dynamics, we quickly realize that the τ_{off} can no longer be taken outside the first moment $E[\tau_{\text{off}}^*]$, and second moment $E[(\tau_{\text{off}}^*)^2]$, or the variance $\text{Var}(\tau_{\text{off}}^*)$ since now it possesses its own distribution. In this case, the variance corresponds to that of a product of two distributions:

$$\text{Var}(\tau_{\text{off}}^*) = \text{Var}(\tau_{\text{off}} \cdot G) = \text{Var}(\tau_{\text{off}}) \text{Var}(G) + \text{Var}(\tau_{\text{off}}) E[G]^2 + \text{Var}(G) E[\tau_{\text{off}}]^2$$

As shown in Supplementary Note 1, the $\text{Var}(\tau_{\text{off}})$ in the glassy adhesion regime diverges, which means that $\text{Var}(\tau_{\text{off}}^*)$ is no longer mathematically defined. The reviewers noted that "...from a modeling standpoint, this is a modest modification: in real cells, any parameter will follow a distribution..." However, this is a critical distinction that goes far beyond simply assigning a distribution to a model parameter. The breakdown of finite variance signifies a violation of the central limit theorem, meaning that stochastic averaging no longer holds. This intrinsic departure from Gaussian

behavior is what gives rise to intermittent trapping and anomalous migration trajectories, phenomena that cannot be captured by models relying solely on parameter distributions.

b) Standard motor-clutch models do not capture intermittent trapping regardless of parameter change:

We next compared the migration trajectories generated by the standard motor-clutch model with those from the glassy adhesion model, as 1D trajectories provide a direct visual representation of trapping behavior while preserving the underlying physics. In Fig. R2, the conventional model assumes a constant value of τ_{off} , resulting in no associated probability distribution. The resulting trajectories are uniform and continuous, apart from small fluctuations arising from the stochastic Monte Carlo implementation. Occasional short pauses appear, corresponding to the stall regimes noted by the reviewers, where clutch forces balance across the substrate. Quantifying these waiting periods reveals that their durations follow a Gaussian distribution, with most events occurring within a narrow range of approximately 5–20 s. This confirms that the stalled phases in the standard model follow Gaussian statistics, consistent with the mathematical analysis above.

In contrast, when τ_{off} follows a heavy-tailed, glassy distribution, the resulting trajectories change qualitatively. While rapid load-fail cycles remain, they are now interspersed with extended intervals of no displacement—true trapping events. The distribution of these trapping times remains dominated by short events around 20 s but now exhibits rare, long-lived pauses reaching up to ~1600 s within a 2000 s trajectory, as shown in the inset. These events occur intermittently and unpredictably, reproducing the heterogeneous dynamics observed in the experimental trajectories. Together, these results illustrate that intermittent trapping emerges only when non-Gaussian statistics are introduced through glassy adhesion dynamics, which generate both short and long trapping events within a unified physical framework and capture anomaly in the model.

The reviewers also comment that “...more sophisticated standard motor clutch models like that of Bangasser et al. are

Figure R2 | Comparison of migration dynamics in conventional and glassy motor-clutch models.

The conventional model (left) with constant τ_{off} produces uniform, Gaussian trajectories with narrowly distributed trapping times. In contrast, the glassy model (right) with a heavy-tailed τ_{off} distribution yields intermittent trapping and non-Gaussian trapping-time statistics with rare, long-lived events.

capable of generating a spectrum of migration regimes, including both sub and super diffusive behavior, when model parameter space is examined more broadly.” and that our previous response “...relies on selectively presented MSD curves of that model with unusually large error bars (as in their Fig. R5), and without disclosure of the parameter values used...”

Firstly, the published results from these models do not exhibit, or even imply, sub- or super-diffusive regimes; all reported dynamics are strictly characterized as purely diffusive. Moreover, as we have detailed above, these models fundamentally cannot produce anomalous regimes due to their Gaussian nature. To further show this point, we had tested the Bangasser et al., Nat. Commun., 2017 [14] model in our last response, using the original code

(<https://oddelab.umn.edu/software>)

provided by the authors and the default parameters. The error bars produced were also the consequence of the default code used as is, which can also be seen in the original code base example file provided. Consistent with our interpretation, these simulations reveal differences in motility coefficients but not transitions between distinct diffusive regimes.

In addition, incorporating dynamic nucleation of motor-clutch modules complicates the understanding of adhesion-level physics and significantly increases computational costs, requiring several days to produce a single trajectory.

Fig. R3: No change in parameter in conventional models can produce anomalous intermittent trapping. Left: Migration trajectories for different parameter changes are plotted. **Right:** Histogram of trapping times for 50 trajectories shows a Gaussian distribution (blue line).

Furthermore, protrusion nucleation is not a key factor in the system we are analyzing, as detailed in the following paragraph. The difficulty of performing systematic parameter sweeps in such models likely explains the significant variability and broad error margins reported.

To overcome this limitation and explore parameter space, we implemented the same underlying motor-clutch physics without nucleation and showed (Fig. 1) that such conventional models do not produce sub- or super-diffusive behavior. We also explored a wide parameter regime for the standard models in (Fig. 2E) and confirmed that models with $\beta > 3$ consistently yield purely diffusive motion regardless of parameter tuning. We further demonstrate this limitation visually in Fig. R3 by systematically varying key parameters, including clutch number and motor number, examining their impact on trajectory and trapping statistics. While these changes can modulate migration speed and the duration of stalled regimes (waiting/trap time) significantly compared to the default parameter case, the trajectories remain uniform and Gaussian, never exhibiting the long-tailed, non-Gaussian statistics observed in the glassy model (Fig. R2). For transparency, we have also provided a standalone code that allows readers to visualize these trajectories and their corresponding statistics for any chosen parameter set.

Changes in manuscript:

- 1) Added Figures R2 and R3 in the SI as Supplementary Figures 6 and 7.
- 2) Added text in the methods section: This fundamental non-Gaussian character allows the glassy adhesion dynamics to enable the anomalous diffusivity, as shown in SI note 1, and the differences in migration trajectory statistics are shown in SI Figures 6 and 7.
- 3) Added text in section 3.3: The inability of conventional motor-clutch models to reproduce these trapping statistics is further demonstrated in Supplementary Figures 6 and 7. As shown in Supplementary Figure 6, trajectories generated by the standard model with constant clutch off-times produce uniform, Gaussian migration with narrowly distributed trapping durations. Even when model parameters such as clutch number, motor number, or stall duration are systematically varied (Supplementary Fig. 7), the resulting trajectories remain diffusive, showing only changes in average speed or pause length but never exhibiting the long-tailed, non-Gaussian statistics characteristic of anomalous migration. These results confirm that parameter tuning within conventional formulations cannot generate intermittent trapping or anomalous diffusion and that such behaviors emerge only when adhesion glassiness is explicitly incorporated into the model.

Concern#3: Alternative mechanisms causing anomalous diffusion

The reviewers further suggest an alternative mechanism that “...anomalous pauses and trapping times not only can be modulated by motor-clutch ratios but can also arise from additional mechanisms such as protrusion nucleation dynamics.” We respectfully contend that such mechanisms, as implemented in existing models, cannot give rise to anomalous migration. First, this has not been demonstrated in any of the cited studies. Second, as we have shown above, the basis for anomaly lies in the system becoming non-Gaussian, where the variance diverges and the central limit theorem no longer applies. In contrast, the protrusion nucleation dynamics in these models are implemented as purely random processes, and there is no parameter within this framework that can lead to a divergence in variance or a transition to non-Gaussian statistics. Consequently, while such models may exhibit variability in migration speeds, they remain mathematically Gaussian and therefore cannot reproduce true anomalous behavior. Further, in our experiments, the observed differences in diffusivity arise solely from substrate relaxation properties, not from protrusion nucleation differences between the two conditions. As shown in the experimental data (Fig. R1/Fig S7), protrusions are present under both conditions, with the width remaining comparable on fast- and slow-relaxing substrates. Specifically, cells on slow-relaxing substrates, which display sub-diffusive migration, still exhibit numerous clearly defined protrusions, which negates the possibility that the observed sub-diffusion arises from a lack of protrusions, as suggested by the reviewers’ statement that “...when cells do not form clearly defined plasma membrane protrusions, cells typically remain stationary and undergo sub-diffusive behavior...”. While we agree with the general statement that cells lacking protrusions often remain stationary, that scenario does not apply here.

Taken together, these points delineate the limits of prior motor–clutch models. Even with broad parameter exploration, those models remain constrained to either continuous motion with $\alpha \approx 1$ or permanent stalling, without capturing intermittent trapping or transitions between $\alpha < 1$ and $\alpha > 1$. Our glassy adhesion dynamics framework introduces the missing physical ingredient of heterogeneous adhesion lifetimes, which directly explains how substrate relaxation governs both intermittent trapping and step sizes and naturally predicts both sub- and super-diffusive regimes within a single framework, in agreement with experimental observations.

Concern#4: Experimental basis for glassy adhesion dynamics

The reviewers state that the manuscript “...extends prior motor clutch models by introducing a power law distribution of clutch lifetimes...” and that “...from a modeling standpoint, this is a modest modification: in real cells, any parameter will follow a distribution...” concluding that “...simply changing a model parameter to follow a distribution is not sufficient to advance the field...” We appreciate this framing because it allows us to clarify the significance of our contribution.

It is true that many biological parameters vary, but what we introduce here is not an arbitrary spread applied to just any parameter. Conventional motor–clutch models are stochastic by design, yet they consistently yield purely diffusive migration, with variability manifesting only as changes in speed. As we noted in our previous response and detailed in the Supplementary Information, this is because their stochastic dynamics can still be reduced to deterministic mean-field behavior. The clutch off-rate is not simply another parameter in this framework: it is the central kinetic quantity that anchors the cell to the substrate and coordinates the mechanochemical feedback regulating migration. Treating it as a constant, as all prior models have done, severely oversimplifies adhesion dynamics. Importantly, it is not the presence of any distribution that produces anomalous transport. Only long-tailed distributions of lifetimes, well established in the mathematics of anomalous diffusion, generate the intermittency and non-Gaussian statistics required for sub- and super-diffusive behavior.

The reviewers contend that “...without new experimental insight or a fundamentally new mechanistic principle, simply changing a model parameter to follow a distribution is not sufficient to advance the field...” We agree with this standard, and our work provides both. We explicitly draw on a separate literature on adhesion protein dynamics, and we import those insights into the motor–clutch framework while also validating the resulting, cell-level predictions experimentally. In doing so, we do not add an arbitrary spread to a parameter; we translate established protein-level observations [15, 16] into a framework that changes the behavior of the model in a principled way.

For instance, adhesion complexes comprise many proteins, including integrin, talin, vinculin, and zyxin, that unfold and unbind through multiple pathways on different timescales [17-19]. Their conformational energy landscapes are rugged, with low barriers that permit rapid transitions and high barriers that yield rare but long-lived states. Experiments show, for example [19], that talin can populate multiple unfolded conformations under the same load, and that zyxin lifetimes fluctuate with mechanical noise rather than following a fixed time constant. These features are

hallmarks of glassy dynamics: a broad distribution of lifetimes in which rare, long events carry disproportionate weight. Capturing this non-uniformity is not a minor reparameterization. It is a fundamentally different representation of adhesion dynamics that provides the mechanistic lever needed to generate intermittent trapping and the observed transitions between sub- and super-diffusive migration at fixed external conditions.

We thank the reviewers for acknowledging that the study “...is well conducted,” and we respectfully disagree with the view that it represents only a “...modest modification.” What we introduce is a mechanistic principle rooted in protein physics and directly supported by experiments: adhesion clusters exhibit glassy dynamics, and this property fundamentally alters how motor-clutch models predict migration. Even if one were to take a step back and regard our model as a modest extension of existing frameworks, the mechanism it reveals is entirely novel. Specifically, migration dynamics emerge from the interplay between cellular and substrate timescales: slow-relaxing substrates prolong adhesion and trapping, producing sub-diffusive behavior, whereas fast-relaxing substrates shorten adhesion lifetimes and enhance step sizes, giving rise to super-diffusion. By identifying and incorporating this missing physical ingredient, our framework unifies sub-diffusion, super-diffusion, and intermittent trapping within a single adhesion-based mechanism—an integration that, to our knowledge, has not been achieved in prior models.

Minor comment:

While on one hand the detailed response from the Authors is appreciated, it also raises concerns. A 17 page response with many rebuttal points being extremely long relative to what is needed to provide a focused response to a Review comment creates concerns that the authors are trying to respond with volume instead of focused and specific responses. More focused responses would improve the impact of the response.

Our response: We appreciate the reviewer’s concern about the length of our previous response. Given the number and breadth of comments, and the nuanced aspects of both modeling and experiments they touched, we felt it important to explain the details carefully and to show the corresponding manuscript changes, which naturally extended the document. Our goal was never to overwhelm with volume but to ensure clarity. In this revision, we have aimed for sharper focus while still providing sufficient detail to fully resolve the concerns, and we hope this balance between thoroughness and brevity makes our responses more effective.

Given the novelty of the mechanism we introduce, the integration of modeling with experimentally validated predictions, and the relevance for understanding cell motility in complex environments, we believe this work is well-suited for consideration in *Nature Communications*. Our study brings attention to an overlooked aspect of adhesion dynamics and provides a framework that can be used to explore anomalous migration across different biological systems. We hope that the interdisciplinary readership of *Nature Communications*, spanning both physics-based modeling and cell biology, will find these insights timely and relevant.

References:

1. Leijnse, N., Y.F. Barooji, M.R. Arastoo, S.L. Sonder, B. Verhagen, L. Wullkopf, J.T. Erler, S. Semsey, J. Nylandsted, L.B. Oddershede, A. Doostmohammadi, and P.M. Bendix, *Filopodia rotate and coil by actively generating twist in their actin shaft*. Nat Commun, 2022. **13**(1): p. 1636.
2. Leijnse, N., L.B. Oddershede, and P.M. Bendix, *Helical buckling of actin inside filopodia generates traction*. Proc Natl Acad Sci U S A, 2015. **112**(1): p. 136-41.
3. Urbancic, V., R. Butler, B. Richier, M. Peter, J. Mason, F.J. Livesey, C.E. Holt, and J.L. Gallop, *Filopodyan: An open-source pipeline for the analysis of filopodia*. J Cell Biol, 2017. **216**(10): p. 3405-3422.
4. Salas-Vidal, E. and H. Lomeli, *Imaging filopodia dynamics in the mouse blastocyst*. Dev Biol, 2004. **265**(1): p. 75-89.
5. Maiuri, P., J.F. Rupprecht, S. Wieser, V. Rupprecht, O. Benichou, N. Carpi, M. Coppey, S. De Beco, N. Gov, C.P. Heisenberg, C. Lage Crespo, F. Lautenschlaeger, M. Le Berre, A.M. Lennon-Dumenil, M. Raab, H.R. Thiam, M. Piel, M. Sixt, and R. Voituriez, *Actin flows mediate a universal coupling between cell speed and cell persistence*. Cell, 2015. **161**(2): p. 374-86.
6. Rupprecht, V., S. Wieser, A. Callan-Jones, M. Smutny, H. Morita, K. Sako, V. Barone, M. Ritsch-Martel, M. Sixt, R. Voituriez, and C.P. Heisenberg, *Cortical contractility triggers a stochastic switch to fast amoeboid cell motility*. Cell, 2015. **160**(4): p. 673-685.
7. Lavi, I., N. Meunier, R. Voituriez, and J. Casademunt, *Motility and morphodynamics of confined cells*. Phys Rev E, 2020. **101**(2-1): p. 022404.

8. Hennig, K., I. Wang, P. Moreau, L. Valon, S. DeBeco, M. Coppey, Y.A. Miroshnikova, C. Albiges-Rizo, C. Favard, R. Voituriez, and M. Baland, *Stick-slip dynamics of cell adhesion triggers spontaneous symmetry breaking and directional migration of mesenchymal cells on one-dimensional lines*. *Sci Adv*, 2020. **6**(1): p. eaau5670.
9. d'Alessandro, J., A. Barbier-Chebbah, V. Cellerin, O. Benichou, R.M. Mege, R. Voituriez, and B. Ladoux, *Cell migration guided by long-lived spatial memory*. *Nat Commun*, 2021. **12**(1): p. 4118.
10. Condamin, S., V. Tejedor, R. Voituriez, O. Benichou, and J. Klafter, *Probing microscopic origins of confined subdiffusion by first-passage observables*. *Proc Natl Acad Sci U S A*, 2008. **105**(15): p. 5675-80.
11. Huda, S., B. Weigelin, K. Wolf, K.V. Tretiakov, K. Polev, G. Wilk, M. Iwasa, F.S. Emami, J.W. Narojczyk, M. Banaszak, S. Soh, D. Pilans, A. Vahid, M. Makurath, P. Friedl, G.G. Borisy, K. Kandere-Grzybowska, and B.A. Grzybowski, *Levy-like movement patterns of metastatic cancer cells revealed in microfabricated systems and implicated in vivo*. *Nat Commun*, 2018. **9**(1): p. 4539.
12. Dieterich, P., R. Klages, R. Preuss, and A. Schwab, *Anomalous dynamics of cell migration*. *Proc Natl Acad Sci U S A*, 2008. **105**(2): p. 459-63.
13. Bangasser, B.L. and D.J. Odde, *Master equation-based analysis of a motor-clutch model for cell traction force*. *Cell Mol Bioeng*, 2013. **6**(4): p. 449-459.
14. Bangasser, B.L., G.A. Shamsan, C.E. Chan, K.N. Opoku, E. Tuzel, B.W. Schlichtmann, J.A. Kasim, B.J. Fuller, B.R. McCullough, S.S. Rosenfeld, and D.J. Odde, *Shifting the optimal stiffness for cell migration*. *Nat Commun*, 2017. **8**: p. 15313.
15. Young, R.D., *Glassy Dynamics in Proteins*. 1989.
16. Satija, R., A.M. Berezhkovskii, and D.E. Makarov, *Broad distributions of transition-path times are fingerprints of multidimensionality of the underlying free energy landscapes*. *Proc Natl Acad Sci U S A*, 2020. **117**(44): p. 27116-27123.
17. Kanchanawong, P., G. Shtengel, A.M. Pasapera, E.B. Ramko, M.W. Davidson, H.F. Hess, and C.M. Waterman, *Nanoscale architecture of integrin-based cell adhesions*. *Nature*, 2010. **468**(7323): p. 580-4.
18. Lele, T.P., J. Pendse, S. Kumar, M. Salanga, J. Karavitis, and D.E. Ingber, *Mechanical forces alter zyxin unbinding kinetics within focal adhesions of living cells*. *J Cell Physiol*, 2006. **207**(1): p. 187-94.
19. Yao, M., B.T. Goult, B. Klapholz, X. Hu, C.P. Toseland, Y. Guo, P. Cong, M.P. Sheetz, and J. Yan, *The mechanical response of talin*. *Nat Commun*, 2016. **7**: p. 11966.

Response to Reviewers.

(The reviewer comments are in blue, our response in plain text, and the edits made to the manuscript are highlighted in yellow)

Reviewers 1 & 2:

The authors have addressed our previous concerns and improved their manuscript.

Our Response: We sincerely thank the reviewers for their positive assessment and constructive feedback throughout the review process. We are glad that the revised manuscript has addressed all concerns, and we appreciate their acknowledgment of the improvements made.

Reviewers 3 & 4:

While this review still has some issues regarding the novelty of the findings, they are technically strong and will make a considerable contribution to the field. It is my recommendation that the manuscript be accepted for publication.

Remarks by Reviewer#3 on code availability: No concerns

Remarks by Reviewer#4 on code availability: I was able to successfully install and run the code without any major issues. Although I did not attempt to reproduce the specific results reported by the authors, the code appears to be well-structured and functional, suggesting that it could serve as a valuable resource for the research community. Additionally, the code includes a comprehensive README file that provides clear and concise instructions for installation and use, which should make it accessible even to users who are not deeply familiar with the underlying methods. Overall, the code seems well-documented and user-friendly, making it a potentially useful tool for others interested in building upon or extending the work presented in the paper.

Our response: We sincerely thank Reviewers 3 and 4 for their thoughtful assessments of our manuscript. We are grateful for their recognition of the technical strengths of our work and its potential contribution to the field. We also appreciate their positive feedback regarding the availability, structure, and usability of our code. Their comments have helped strengthen and clarify the manuscript, and we are pleased that their concerns were fully resolved in the revised version.